# Selective Prompt Anchoring for Code Generation

**Yuan Tian** [1]    **Tianyi Zhang** [1]

## Abstract

Recent advances in large language models (LLMs) have transformed software development by automatically generating code from natural language. Yet challenges remain in generating fully correct code that aligns with user intent. Our study reveals that LLMs tend to pay less attention to user prompts as more code tokens are generated. We hypothesize that this attention dilution issue is an important reason for code generation errors. To mitigate this issue, we propose **S**elective **P**rompt **A**nchoring (SPA) to guide code LLMs to pay more attention to user intent when generating code. We evaluate SPA using six base LLMs across six benchmarks. Our results demonstrate that SPA enhances Pass@1 by up to 12.9%, consistently outperforming SOTA methods in all settings. Our code is available at https://github.com/magic-YuanTian/Selective-Prompt-Anchoring.

## 1. Introduction

Large language models (LLMs) have emerged as powerful programming assistants. They have demonstrated unprecedented capabilities in interpreting natural language descriptions of programming tasks and generating source code (Guo et al., 2024; Rozière et al., 2024). Despite this great progress, LLMs still produce incorrect solutions to challenging tasks or generate code that does not fully meet user expectations (Wang et al., 2024; Ning et al., 2023).

To improve the performance of LLMs on coding tasks, many efforts have been made to develop high-quality training data (Li et al., 2023; Guo et al., 2024; Wei et al., 2023) and design new domain-specific training objectives (Niu et al., 2022; Chakraborty et al., 2022). However, these approaches require tremendous computational resources. To address

this challenge, various prompting methods have been explored to enhance the inference process of code LLMs, e.g., retrieval-augmented generation (Du et al., 2024), chain-of-thoughts (Le et al., 2024; Suzgun et al., 2022), self-planning and debugging (Jiang et al., 2024; Chen et al., 2024), etc.

Despite these efforts, little is known about why LLMs fail to generate correct code. In this work, we seek to bridge the knowledge gap by investigating the attention pattern of LLMs during code generation. We analyzed five code LLMs and found that as more code tokens were generated, LLMs paid less attention to the user prompt. This caused LLMs to gradually deviate from the user intent, thereby leading to code generation errors. Furthermore, we found that more generated tokens led to worse code generation performance, demonstrating their struggle with long-term attention.

To mitigate this limitation, we propose **S**elective **P**rompt **A**nchoring (SPA), a model-agnostic approach that optimizes LLMs' attention by amplifying the contextual impact of the user prompt. SPA is inspired by the anchoring effect (Furnham & Boo, 2011) in psychology, which refers to the phenomenon where humans can be influenced by specific information provided before decision-making. In SPA, we refer to this information as *anchored text*, a group of selected prompt tokens that should receive higher attention from the model than others. Figure 1 illustrates the workflow of SPA. Given the anchored text, SPA creates an original embedding matrix (①) as well as a masked embedding matrix by replacing the embeddings corresponding to anchored text with mask embeddings (②). We mathematically show that the anchored text's contextual impact can be approximated by the difference between the logit distribution generated from the original prompt and the prompt with the anchored text masked (③). To amplify the impact of anchored text during code generation, SPA multiplies this logit distribution difference by a hyperparameter called *anchoring strength* (④), and then adds it to the original logit distribution (⑤).

We evaluate SPA on six benchmarks using six code LLMs. The benchmarks cover different programming languages and task difficulty levels, while the LLMs vary in size and code generation performance. SPA enhances Pass@1 by up to 12.9% across all settings, outperforming four SOTA prompting methods and one SOTA attention steering method. Notably, with SPA, a smaller version of DeepSeek-

[1]Department of Computer Science, Purdue University, West Lafayette, IN, USA. Correspondence to: Yuan Tian <tian211@purdue.edu>, Tianyi Zhang <tianyi@purdue.edu>.

*Proceedings of the 42nd International Conference on Machine Learning*, Vancouver, Canada. PMLR 267, 2025. Copyright 2025 by the author(s).

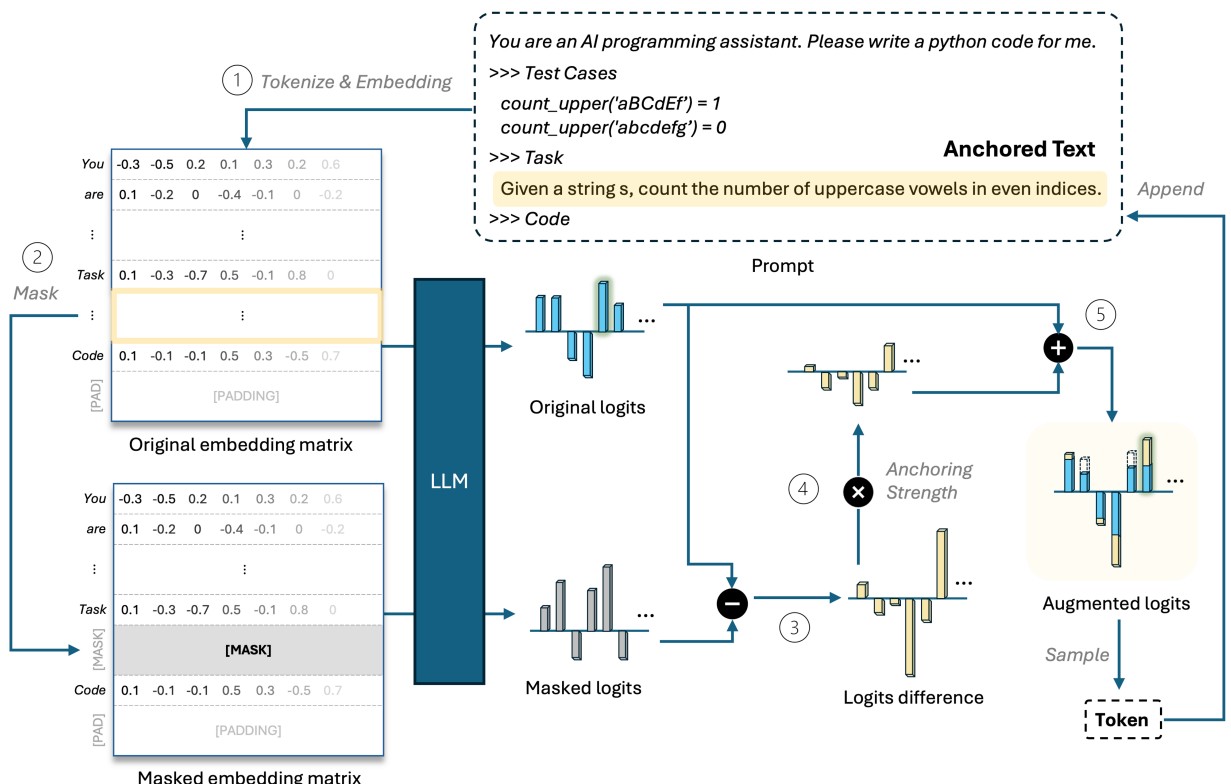

*Figure 1.* The Workflow of Selective Prompt Anchoring (SPA).

Coder (6.7B) can outperform its larger counterpart (33B).

## 2. Attention Analysis of Code LLMs

We conduct an empirical study to investigate the attention dilution phenomenon in code LLMs. Following prior studies (Zhang et al., 2022; Galassi et al., 2021), we obtain self-attention scores from the last layer in LLMs, which has been shown to represent the most accurate attention distribution (Kou et al., 2024; Wan et al., 2022a).[1] We calculate the percentage of attention on the user prompt. Calculation details are provided in Appendix B.4. We also experimented with an alternative gradient-based attention calculation method (Selvaraju et al., 2016) and obtained similar results, as detailed in Appendix B.2.

We analyzed five code LLMs on HumanEval (Chen et al., 2021) and LiveCodeBench (Jain et al., 2024). Figure 2 shows the shift of LLMs' attention on the user prompt dur-

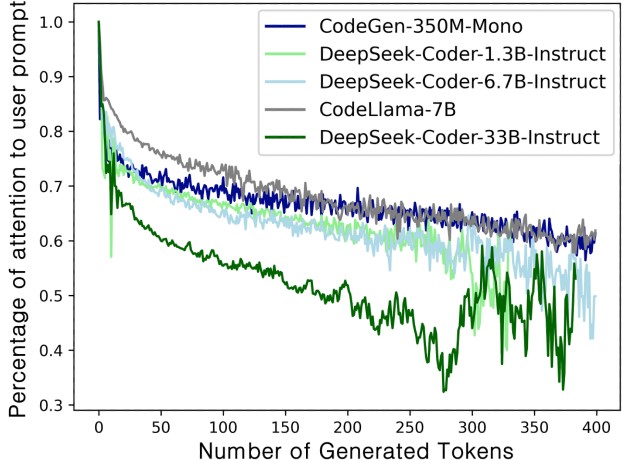

*Figure 2.* Shift of LLMs' self-attention to the user prompt.

ing code generation. The result shows that as more tokens are generated, models' attention on the user prompt gradually becomes less. Consequently, the code generation pro-

---

[1]Intuitively, deeper layers capture representations with long-distance dependencies such as the control flow, which mirrors how humans understand programs.

*Table 1.* Length of correct vs. incorrect code generated by LLMs.

|  | EASY | MEDIUM | HARD | OVERALL |
|---|---|---|---|---|
| PASSED | 294 | 475 | 400 | 390 |
| FAILED | 418 | 664 | 784 | 622 |

cess becomes increasingly influenced by tokens generated in recent time steps, rather than the user prompt. This can be problematic in two ways. First, any generation inaccuracy, such as creating an unhelpful variable in earlier steps, is likely to propagate and influence subsequent steps. Second, when generating long and complex code with many detailed requirements, the model may overlook some specifications in the early stages, as the model tends to focus more on recent tokens. To further investigate whether this phenomenon affects performance, we experimented on Live-CodeBench (Jain et al., 2024), which provides 3 difficulty levels for each task. Table 1 shows that, for tasks with the same difficulty level, the average length of incorrectly generated code is consistently longer than that of correctly generated code. This implies that more generation steps can indeed cause attention dilution and hinder accuracy. More details are included in the Appendix A.

# 3. Approach

## 3.1. Overview

Given a user prompt $x$, a code LLM $f_\theta$ generates tokens autoregressively. At step $i$, the input to $f_\theta$ is an $n \times m$ embedding matrix $\mathbf{E}_i$, defined as:

$$\mathbf{E}_i = [\mathbf{E}^x, e_1, e_2, \ldots, e_{i-1}, \texttt{PAD}]. \tag{1}$$

where $\mathbf{E}^x$ is the submatrix of embeddings for tokens in user prompt $x$, the series $e_1, \ldots, e_{i-1}$ are embeddings of generated tokens, and PAD is a padding submatrix. The model outputs logits and transforms them into a probability distribution. Then, a sampling method is applied to select the next token.

We propose Selective Prompt Anchoring (SPA) to steer model attention by amplifying the contextual impact of the important tokens in the user prompt. In this work, we assume these important tokens are selected by users. Inspired by the anchoring effect (Furnham & Boo, 2011) in psychology, we call these important tokens "*anchored text*". For example, in the prompt, "*Given a string, count the number of uppercase vowels*", the user may want to emphasize "*uppercase vowels*" to ensure that the model does not forget to define uppercase vowels (i.e., "*A, E, I, O, U*") in the generated code.

In Section 3.2, we first mathematically model how to steer model attention. Then we augment model attention by in-

creasing the impact of user intent and derive the augmented model output. In Section 3.3, we derive and calculate the augmented logits, which can be approximately represented by the linear combination of original and masked logits.

## 3.2. Attention Steering and Prompt Anchoring

SPA performs attention steering by scaling the impact of selected tokens to the output logits $f_\theta(\mathbf{E}_i)$. Suppose $x$ is the *anchored text* for which we want to adjust the impact. $\mathbf{E}_i$ is an $n \times m$ input embedding matrix at step $i$, and $\mathbf{E}^x$ represents a $n \times k$ submatrix within $\mathbf{E}_i$ corresponding to the $x$. They are visualized below:

$$\mathbf{E}_i = \begin{bmatrix} e_{11} & \cdots & e_{1k} & e_{1,k+1} & \cdots & e_{1m} \\ e_{21} & \cdots & e_{2k} & e_{2,k+1} & \cdots & e_{2m} \\ \vdots & \ddots & \vdots & \vdots & \ddots & \vdots \\ e_{n1} & \cdots & \underbrace{e_{nk}}_{\mathbf{E}^x} & e_{n,k+1} & \cdots & e_{nm} \end{bmatrix}. \tag{2}$$

We construct two $n \times m$ matrices, $\mathbf{X}$ for user prompt and $\mathbf{G}_i$ for generated code.

- To construct $\mathbf{X}$, we only retain $k$ columns of $\mathbf{E}_i$ that correspond to $\mathbf{E}^x$, while setting remaining columns to zero. This matrix $\mathbf{X}$ remains constant during token generation.

- Conversely, $\mathbf{G}_i$ is formed by zeroing out the same $k$ columns of $\mathbf{E}_i$ that correspond to $\mathbf{E}^x$, while preserving all elements in the remaining columns.

They are visualized as follows:

$$\mathbf{X} = \begin{bmatrix} e_{11} & e_{12} & \cdots & e_{1k} & 0 & \cdots & 0 \\ e_{21} & e_{22} & \cdots & e_{2k} & 0 & \cdots & 0 \\ \vdots & \vdots & \ddots & \vdots & \vdots & \ddots & \vdots \\ e_{n1} & e_{n2} & \cdots & e_{nk} & 0 & \cdots & 0 \end{bmatrix}, \tag{3}$$

$$\mathbf{G}_i = \begin{bmatrix} 0 & 0 & \cdots & 0 & e_{1,k+1} & \cdots & e_{1m} \\ 0 & 0 & \cdots & 0 & e_{2,k+1} & \cdots & e_{2m} \\ \vdots & \vdots & \ddots & \vdots & \vdots & \ddots & \vdots \\ 0 & 0 & \cdots & 0 & e_{n,k+1} & \cdots & e_{nm} \end{bmatrix}. \tag{4}$$

$\mathbf{X}$ encapsulates the semantics for the anchored text $x$, while $\mathbf{G}_i$ encapsulates the semantics for the remaining text, such as the generated code. The sum of $\mathbf{X}$ and $\mathbf{G}_i$ reconstructs the original matrix $\mathbf{E}_i$:

$$\mathbf{E}_i = \mathbf{X} + \mathbf{G}_i. \tag{5}$$

Suppose we want to scale the semantic impact of the matrix

$\mathbf{X}$ by a value $\omega$.[2] We refer to $\omega$ as *anchoring strength*. We use $\mathbf{E}_i(\mathbf{X}, \omega)$ to represent the function that augments $\mathbf{E}_i$ by scaling the influence of $\mathbf{X}$ by a degree of $\omega$ in the final logit.

- $\omega > 1$ indicates semantic amplification, meaning the model generates code with greater consideration of $\mathbf{X}$.
- $\omega = 1$ indicates using the original embedding. $\mathbf{E}_i$ is equivalent to $\mathbf{E}_i(\mathbf{X}, 1)$.
- $0 \le \omega < 1$ indicates semantic diminishment, meaning the model generates code with less consideration of $\mathbf{X}$. When $\omega$ is 0, the model does not consider $\mathbf{X}$ at all.
- $\omega < 0$ indicates a reversed semantic impact, meaning the model generates the code in the opposite manner. For example, if $\mathbf{X}$ corresponds to *"uppercase"*, the model will instead consider *"lowercase"*.

In this work, we focus on scaling up the impact of the anchored text $x$ to mitigate the attention dilution issue. Let $F_{\theta,i,x}(\omega)$ represent the augmented logits calculated by model $f_\theta$ at step $i$, where the impact of anchored text $x$ is scaled by $\omega$. We can calculate the integral of the partial derivative of $f_\theta$ with respect to $\omega$ from 0 to $\omega$.[3] Formally,

$$F_{\theta,i,x}(\omega) = f_\theta(\mathbf{E}_i(\mathbf{X}, \omega) + \mathbf{G}_i) \qquad (6)$$

$$= F_{\theta,i,x}(0) + \int_0^\omega \frac{dF_{\theta,i,x}(t)}{dt}\, dt, \qquad (7)$$

where $t$ is the variable of integration.

To reduce computational overhead, attention augmentation is activated when the original generation fails the test case, which is typically included in the code generation context. When no test case is available, attention augmentation is applied continuously throughout the generation. In the following section, we explain how to calculate the augmented logits through approximation.

### 3.3. Augmented Logits by Approximation

Given the computational complexities of LLMs, directly solving $\int_0^\omega \frac{dF_{\theta,i,x}(t)}{dt}\, dt$ in Equation 7 is impractical. Therefore, we approximate it by employing the Taylor expansion:

$$F_{\theta,i,x}(\omega) = F_{\theta,i,x}(0) + \omega \cdot F_{\theta,i,x}{}'(0) + \frac{\omega^2}{2!} F_{\theta,i,x}{}''(0) + \dots \tag{8}$$

Since LLMs are inherently non-linear, higher-order derivatives of the logits function are non-zero. We truncate the

Taylor expansion in Equation 8 after the first derivative to obtain an approximation, yielding:

$$F_{\theta,i,x}(\omega) \approx F_{\theta,i,x}(0) + \omega \cdot F_{\theta,i,x}{}'(0), \qquad (9)$$

where the the integral part $\int_0^\omega \frac{dF_{\theta,i,x}(t)}{dt}\, dt$ in Equation 7 is approximated by $\omega \cdot F_{\theta,i,x}{}'(0)$. To calculate $F_{\theta,i,x}(0)$,[4] we mask the anchored text $x$ using mask embeddings. Each LLM provides special tokens reserved for text masking, which almost has no semantic impact, e.g., `<unk>` for Code Llama (Rozière et al., 2024) and `<pad>` for DeepSeek-Coder (Guo et al., 2024). Each special token corresponds to a mask embedding. By replacing embeddings of $x$ with mask embeddings, we get a masked input matrix $\mathbf{E}_i^{mask}$. It ablates the semantic impact of the anchored text $x$ while ensuring that the positional encoding remains unaffected. Thus, we can get

$$F_{\theta,i,x}(0) = f_\theta(\mathbf{E}_i^{mask}). \qquad (10)$$

To calculate $F_{\theta,i,x}{}'(0)$, we use finite-difference methods to get an approximation. Assuming the interval of $1 - 0$ is sufficiently small for $F_{\theta,i,x}$, we get:

$$F_{\theta,i,x}{}'(0) \approx \frac{F_{\theta,i,x}(1) - F_{\theta,i,x}(0)}{1 - 0}. \qquad (11)$$

Combining Equations 9, 12 and 13, we get the augmented logits by first-order approximation:

$$F_{\theta,i,x}(\omega) \approx F_{\theta,i,x}(0) + \omega \cdot (F_{\theta,i,x}(1) - F_{\theta,i,x}(0)) \quad (12)$$

$$= \omega \cdot f_\theta(\mathbf{E}_i(\mathbf{X}, 1) + \mathbf{G}_i)$$

$$+ (1 - \omega) \cdot f_\theta(\mathbf{E}_i(\mathbf{X}, 0) + \mathbf{G}_i) \qquad (13)$$

$$= \omega \cdot f_\theta(\mathbf{E}_i) + (1 - \omega) \cdot f_\theta(\mathbf{E}_i^{mask}). \qquad (14)$$

Based on the augmented logits $F_{\theta,i,x}(\omega)$ where the impact of the anchored text is scaled by the value $\omega$, a certain sampling algorithm (e.g., greedy sampling) can be applied to select the token. We provide more discussion about approximation in Appendix C.

We choose not to directly modify self-attention layers to steer the model attention, since this requires identifying which attention head in which layer to steer and the direct editing of attention values does not synchronize with other components like the feedforward layers, which can be fairly brittle and costly. Instead, we chose to simulate attention steering by logits manipulation, which is fast, reliable, and model-agnostic. In the next section, we demonstrate that SPA achieves better performance with less computational overhead compared to a SOTA method that directly modifies self-attention layers (Zhang et al., 2024).

---

[2] Scaling the semantics of $\mathbf{X}$ by $\omega$ is not equivalent to multiplying $\mathbf{X}$ by $\omega$. Multiplying the embedding of $\mathbf{X}$ by $\omega$ does not simply improve the "semantic influence" of $\mathbf{X}$, since the embedding of $\mathbf{X}$ also encodes other non-semantic information such as positional information. This is why we compute the difference between the logits when masking and unmasking $\mathbf{X}$ to cancel out noise and some of the non-semantic information (detailed in Section 3.3).

[3] $f_\theta$ is differentiable for backpropagation.

[4] $F_{\theta,i,x}(0)$ does not mean setting the embedding vector to zeros. Instead, it means setting $\omega$ to zero, which replaces the original embedding for anchored text with the mask embedding that contains no semantic information. This mask embedding vector is non-zero. Conversely, a zero embedding does not necessarily indicate the absence of semantic information.

# 4. Experiment Setup

## 4.1. Benchmarks

**HumanEval** (Chen et al., 2021) includes 164 Python tasks designed by OpenAI and now has become a widely-used benchmark for code generation.

**MBPP** (Austin et al., 2021) is another popular benchmark that includes 974 crowd-sourced Python tasks. Due to ambiguous task descriptions, the authors of MBPP created a sanitized version that included 427 tasks with clearer descriptions. We evaluate SPA on the sanitized version.

**HumanEval+** and **MBPP+** (Liu et al., 2023) improves the original HumanEval and MBPP benchmarks with additional test cases to cover corner cases (Liu et al., 2024b).

**HumanEval-X** (Hendrycks et al., 2021a) extends the HumanEval benchmark to support more programming languages such as Python, Java, JavaScript, C++, and Go. It aims to evaluate the multilingual code generation abilities.

**BigCodeBench** (Zhuo et al., 2024) is a more challenging benchmark for code generation that evaluates models' abilities to follow complex instructions and use tools, including 1,140 real-world Python tasks across 139 libraries.

**LiveCodeBench** (Jain et al., 2024) is a contamination-free benchmark sourced from competitive programming platforms. The benchmark continues to evolve and add new code generation tasks. We conducted experiments on the latest release, which includes a total of 1,055 tasks. The latest tasks span from October 1, 2024, to February 1, 2025.

## 4.2. Baselines

**Base Models.** We select six representative open-source code LLMs: CodeGen-Mono-350M (Nijkamp et al., 2023), CodeLlama-7B (Rozière et al., 2024), StarCoder2-15B (Lozhkov et al., 2024), and DeepSeek-Coder-Instruct-1.3B, 6.7B, and 33B (Guo et al., 2024). These models exhibit varying levels of performance in code generation. We include more setup details about models in Appendix H.

**Attention Steering Baseline.** PASTA (Zhang et al., 2024) is a recent method designed to steer model attention for better instruction following. Unlike SPA, PASTA requires a model-specific and time-consuming profiling stage to identify attention headers that are beneficial to performance. For each unique task, PASTA requires around 1000 training samples to identify attention headers that are beneficial to performance. By contrast, SPA is model-agnostic and only needs tuning a single hyperparameter with a few samples, which is fast and generalizable as detailed in Appendix J. PASTA internally edits transformers' self-attention during the feed-forward process, whereas SPA only edits the final logits based on a mathematical approximation.

**Prompting Methods.** In addition to PASTA, we also compare SPA to the mainstream prompting-based code generation methods, including Self-Debugging (Chen et al., 2024), Self-Planning (Jiang et al., 2024), ReAct (Yao et al., 2023), and Self-Edit (Zhang et al., 2023b). Self-Debugging and Self-Edit leverage error messages from test cases to refine generated code. Self-planning generates a step-by-step plan before code generation. ReAct prompts an LLM to generate reasoning traces and action plans in an interleaved manner.

## 4.3. Evaluation Metrics

Following prior work (Chen et al., 2021), we measure code generation performance using the Pass@$k$ metric, which measures whether any of the top $k$ candidates can pass all the test cases. For main results, we report Pass@1 using greedy sampling to generate a single code snippet. To demonstrate the generalizability of SPA, we further calculate Pass@10 using beam search in Appendix F.

## 4.4. Anchoring Strength Tuning

The anchoring strength $\omega$ serves as a hyperparameter in SPA. For each model and dataset, we use grid search to tune the anchoring strength $\omega$ on 1/5 of the tasks, and evaluate Pass@1 of SPA on the remaining 4/5 of the tasks. This process is repeated across all five folds, with final performance metrics averaged across folds.[5] We observe an unimodal relationship between $\omega$ and the performance. We provide more details for the tuning SPA in Appendix J.

# 5. Results

## 5.1. Improvement over Base Models

Table 2 shows SPA consistently improves Pass@1 across all 6 benchmarks and 6 code LLMs. On HumanEval/HumanEval+ and MBPP/MBPP+, SPA enhances the base model with on average 5.5% absolute improvement and 14.5% relative improvement, achieving up to a 12.9% absolute improvement and a 42% relative improvement. These improvements are observed across models of varying sizes (350M-33B), original performance (15%-86%), and architectures. Notably, with SPA, the smaller DeepSeek-Coder (6.7B) outperforms its much larger 33B counterpart on HumanEval. This suggests optimizing model attention is a promising alternative to simply scaling up model size (Kaplan et al., 2020) for performance improvement. On BigCodeBench and LiveCodeBench, while the absolute improvements (average 1.57% and 1.90%) are more modest compared to other benchmarks, the relative gains (average 22.83% and 28.50%) remain significant. This is because

---

[5]We tune SPA on all tasks for better generalizability, while SPA is only activated for failed tasks during inference to optimize computational efficiency and generation performance.

*Table 2.* Absolute ($\Delta$) and Relative ($\uparrow$) Performance improvements in Pass@1 rates (%).

| MODEL | SIZE | HUMANEVAL | HUMANEVAL+ | MBPP | MBPP+ | BIGCODEBENCH | LIVECODEBENCH |
|---|---|---|---|---|---|---|---|
| CODEGEN-MONO | (350M) | 15.3 | 12.2 | 19.6 | 15.9 | 1.1 | 1.1 |
| + SPA | | 20.1 $\Delta+4.8$ (31% $\uparrow$) | 17.1 $\Delta+4.9$ (40% $\uparrow$) | 27.4 $\Delta+7.8$ (40% $\uparrow$) | 22.6 $\Delta+6.7$ (42% $\uparrow$) | 1.6 $\Delta+0.5$ (45% $\uparrow$) | 1.5 $\Delta+0.4$ (36% $\uparrow$) |
| DEEPSEEK-CODER | (1.3B) | 66.4 | 61.8 | 58.2 | 52.4 | 2.5 | 6.5 |
| + SPA | | 70.1 $\Delta+3.7$ (6% $\uparrow$) | 67.7 $\Delta+5.9$ (10% $\uparrow$) | 60.9 $\Delta+2.7$ (5% $\uparrow$) | 52.4 $\Delta+0.0$ (0% $\uparrow$) | 3.4 $\Delta+0.9$ (36% $\uparrow$) | 9.2 $\Delta+2.7$ (42% $\uparrow$) |
| DEEPSEEK-CODER | (6.7B) | 75.6 | 70.2 | 67.0 | 58.5 | 12.7 | 7.8 |
| + SPA | | 88.5 $\Delta+12.9$ (17% $\uparrow$) | 79.9 $\Delta+9.7$ (14% $\uparrow$) | 71.0 $\Delta+4.0$ (6% $\uparrow$) | 60.7 $\Delta+2.2$ (4% $\uparrow$) | 16.4 $\Delta+3.7$ (29% $\uparrow$) | 10.8 $\Delta+3.0$ (38% $\uparrow$) |
| CODELLAMA | (7B) | 33.6 | 28.2 | 50.9 | 40.8 | 3.4 | 3.8 |
| + SPA | | 44.0 $\Delta+10.4$ (31% $\uparrow$) | 36.0 $\Delta+7.8$ (28% $\uparrow$) | 54.3 $\Delta+3.4$ (7% $\uparrow$) | 44.0 $\Delta+3.2$ (8% $\uparrow$) | 4.1 $\Delta+0.7$ (5% $\uparrow$) | 4.0 $\Delta+0.2$ (5% $\uparrow$) |
| STARCODER2 | (16B) | 67.7 | 60.4 | 78.0 | 65.1 | 13.3 | 7.0 |
| + SPA | | 75.6 $\Delta+7.9$ (12% $\uparrow$) | 65.6 $\Delta+5.2$ (9% $\uparrow$) | 82.0 $\Delta+4.0$ (5% $\uparrow$) | 69.1 $\Delta+4.0$ (6% $\uparrow$) | 14.3 $\Delta+1.0$ (8% $\uparrow$) | 8.2 $\Delta+1.2$ (17% $\uparrow$) |
| DEEPSEEK-CODER | (33B) | 81.7 | 77.1 | 73.4 | 63.2 | 18.9 | 11.9 |
| + SPA | | 86.2 $\Delta+4.5$ (6% $\uparrow$) | 79.3 $\Delta+2.2$ (3% $\uparrow$) | 79.4 $\Delta+6.0$ (8% $\uparrow$) | 70.3 $\Delta+7.1$ (11% $\uparrow$) | 21.5 $\Delta+2.6$ (14% $\uparrow$) | 15.8 $\Delta+3.9$ (33% $\uparrow$) |

SPA leverages the ability of its base model. If the base model could solve a task but overlooks a few important tokens in the prompt, SPA can help with this by adjusting the attention. If a model lacks the ability to solve a task, adjusting the model attention will not help much.

To demonstrate the generalizability of SPA, we show that SPA can consistently improve Pass@10 by, on average 3.42% and up to 7.9%, as detailed in Appendix F. We further evaluate SPA in scenarios where test cases are not available and show that it still significantly enhances performance, achieving an average 4.7% Pass@1 improvement on HumanEval, as detailed in Appendix K. We illustrate SPA's attention anchoring with two concrete examples in Appendix D. SPA improves performance by simply aligning attention to prompts, without introducing additional model parameters or context. We believe this success comes from its ability to improve attention reliability. We provide a thorough discussion in Appendix G.

## 5.2. Comparison to SOTA Methods

**PASTA.** Table 3 shows that SPA outperforms PASTA by achieving a 5.9X higher Pass@1 improvement while using only 20% of the inference time.[6] We include more experimental details and discussion in Appendix L. Compared to the base model, SPA increases the decoding time by a practically negligible factor of 1.27. Further discussion about the computational cost of SPA can be found in Appendix E.

**Prompting Methods.** Table 3 shows that SPA outperforms Self-Debugging, Self-Edit, Self-Planning, and ReAct by achieving improvements that are 1.8X, 4.3X, 2.1X, and 5.9X higher, respectively, while only using 36%, 37%, 45%,

and 34% of the inference time. The performance superiority stems from two key factors. On the one hand, unlike prompting methods that add more information or enforce generation workflows, SPA preserves the original prompt and does not involve additional LLM calls, thereby achieving better time efficiency. On the other hand, SPA addresses the attention dilution issue that prompting methods do not explicitly handle, thereby achieving higher accuracy.

*Table 3.* Comparison between SPA and SOTA methods.

| METHOD | $\Delta$PASS@1 (%) | TIME (SEC) |
|---|---|---|
| BASE MODEL | 0 | 7.7 |
| PASTA | +1.2 | 48.8 |
| SELF-DEBUGGING | +4.2 | 27.3 |
| SELF-EDIT | +1.8 | 26.4 |
| SELF-PLANNING | +3.6 | 21.6 |
| REACT | +1.3 | 28.8 |
| SPA | **+7.7** | **9.8** |

## 5.3. Evaluation on Different Programming Languages

To evaluate the generalizability across different programming languages, we further evaluate SPA using HumanEval-X (Hendrycks et al., 2021a), which includes five programming languages.[7] Table 4 demonstrates that SPA consistently improves Pass@1 on HumanEval-X, with an average increase of 7.9% for Python, 4.85% for Java, 6.5% for JavaScript, 3.65% for C++, and 5.2% for Go.

## 5.4. Ablation Study of Anchored Text Selection

To investigate the impact of anchored text selection in code generation tasks, we conduct an ablation study by masking

---

[6]For a fair comparison, we average and add the latency of PASTA's model profiling and the SPA's tuning to the inference time for each task.

[7]In the latest version, Rust caused issues when running test cases, so we excluded Rust from the results.

*Table 4.* Evaluation on HumanEval-X with Different Languages.

| MODEL | PY | JAVA | JS | C++ | GO |
|---|---|---|---|---|---|
| CODEGEN (350M) | 15.3 | 9.8 | 13.4 | 9.8 | 6.7 |
| +SPA | +4.8 | +3.0 | +4.3 | +3.7 | +6.7 |
| DEEPSEEK (1.3B) | 66.4 | 42.7 | 57.3 | 43.3 | 40.2 |
| +SPA | +6.7 | +3.0 | +3.0 | +2.4 | +1.8 |
| DEEPSEEK (6.7B) | 75.6 | 48.8 | 65.2 | 49.4 | 45.7 |
| +SPA | +12.9 | +8.5 | +11.6 | +1.8 | +7.3 |
| CODELLAMA (7B) | 33.6 | 22.0 | 29.3 | 22.0 | 20.1 |
| +SPA | +10.4 | +6.1 | +8.5 | +6.1 | +6.7 |
| STARCODER2 (15B) | 67.7 | 22.0 | 29.3 | 22.0 | 20.1 |
| +SPA | +7.9 | +5.5 | +7.9 | +5.5 | +6.1 |
| DEEPSEEK (33B) | 81.7 | 53.0 | 70.7 | 53.7 | 49.4 |
| +SPA | +4.5 | +3.0 | +3.7 | +2.4 | +2.4 |

different components in code generation prompts. We decompose the code generation prompt into three components: (1) *natural language description*, (2) *source code*, and (3) *test cases*. We create 4 automated anchored text selection methods by ablating the source code and test cases.[8] To explore whether anchoring on a few more informative tokens can enhance performance, we further create a condition using important tokens labeled by human programmers as the anchored text. We leverage the dataset from Kou et al. (2024), where multiple programmers manually identified critical tokens that models should attend to when solving HumanEval and MBPP programming tasks. Following our previous experimental setup, we tune SPA for each condition and benchmark, calculating the average Pass@1 improvement across all 6 experimental LLMs.

*Table 5.* Pass@1 Improvement (%) with Different Anchored Texts (HE: HumanEval, MB: MBPP, BCB: BigCodeBench, LCB: LiveCodeBench).

| ANCHORED TEXT | HE | HE+ | MB | MB+ | BCB | LCB |
|---|---|---|---|---|---|---|
| HUMAN ATTENTION (KOU ET AL., 2024) | 3.66 | 3.04 | 2.58 | 2.11 | N/A | N/A |
| NATURAL LANGUAGE | **5.48** | **5.08** | **4.26** | **3.22** | **1.57** | **1.90** |
| + CODE | 4.87 | 4.65 | 3.75 | 2.81 | 1.42 | 1.73 |
| + TEST CASES | 5.11 | 4.89 | 4.05 | 3.11 | 1.50 | 1.82 |
| + CODE & TEST CASES | 4.76 | 4.57 | 3.98 | 2.81 | 1.44 | 1.73 |

As shown in Table 5, anchoring the natural language description alone achieves the best performance improvement. For example, on HumanEval, anchoring the natural language description alone improves Pass@1 by 5.48%. Including source code (4.87%) and test cases (5.11%) in the anchored text make the performance worse. When anchoring the en-

---

[8]We do not ablate the natural language description since it explicitly represents the user intent.

tire user prompt, performance deteriorates further to 4.76%. This suggests that removing less relevant context can help SPA better align LLM attention with the user intent defined in the natural language description.

Interestingly, we find that anchoring using human-labeled important tokens achieves only 3.66% on HumanEval, which is consistently less effective than all the automated experimental anchoring methods. It suggests that narrowing the anchored text to more informative tokens does not necessarily improve performance. We think there are two plausible reasons. First, since LLMs need to attend to different context tokens at each decoding step, providing a narrow set of anchored tokens may have a negative impact and distract the LLM in certain decoding steps. Second, previous studies such as Xiao et al. (2024) show that even though some tokens, such as separators and empty space, may not be semantically meaningful or informative, they provide important signals for LLMs to generate the right content (e.g., following the grammar rules). Thus, over-attending to the informative tokens but not the special tokens in the task description may disrupt the regular generation process. Nevertheless, we think this is a challenging but interesting future direction to investigate. We believe our findings will open up new research opportunities for the community on this topic.

### 5.5. Impact of Prompt Length

To further validate that SPA effectively addresses the attention dilution issue, we analyze both the original model performance and SPA's effectiveness across different prompt lengths, as shown in Table 6. We divide the HumanEval dataset into three equal-sized subsets (*Short*, *Medium*, and *Long*) based on the 33rd and 66th percentiles of prompt lengths. We find that the original code LLMs consistently achieve better performance on shorter prompts compared to longer ones. The average Pass@1 is 74.3% for *Short* prompts, 49.2% for *Medium* prompts, and only 30.8% for *Long* prompts. This result echoes our empirical finding on attention dilution, as longer prompts can lead to more severe attention dilution issues.

We further investigate whether SPA can effectively address the attention dilution issue by comparing performance improvements across three subsets of tasks, *Short*, *Medium*, and *Long*. To ensure a fair comparison, the attention augmentation of SPA is activated for all tasks in each subset of this experiment. The average Pass@1 improvement of SPA is 1.1% for *Short*, 3.2% for *Medium*, and 11.7% for *Long*. It shows that SPA consistently achieves greater improvements for longer prompts compared to shorter ones. This implies that SPA indeed improves performance by addressing the attention dilution issue, as the improvement is more significant when attention dilution is more severe.

*Table 6.* SPA's performance on tasks with different prompt lengths.

| MODEL | SHORT | MEDIUM | LONG |
|---|---|---|---|
| CODEGEN (350M) | 37.0 | 6.6 | 2.3 |
| + SPA | +1.7 | +3.0 | +4.3 |
| DEEPSEEK-CODER (1.3B) | 81.8 | 45.5 | 29.6 |
| + SPA | -1.8 | +14.5 | +26.0 |
| DEEPSEEK-CODER (6.7B) | 87.3 | 69.1 | 44.4 |
| + SPA | +3.6 | -3.6 | +7.5 |
| CODELLAMA (7B) | 69.2 | 43.5 | 0 |
| + SPA | +2.6 | +0.0 | +10.0 |
| STARCODER2 (15B) | 85.5 | 49.1 | 40.1 |
| + SPA | -1.8 | +5.5 | +14.8 |
| DEEPSEEK-CODER (33B) | 85.5 | 81.8 | 68.5 |
| + SPA | +1.8 | +0.0 | +7.4 |

### 5.6. Impact of Anchoring Strength

In this section, we discuss how the anchoring strength $\omega$ of SPA influences performance. We observe that different values of $\omega$ yield different effects: when $\omega$ is too low, performance gains are limited; when it is too high, the LLM becomes biased, leading to performance degradation. As shown in Figure 3, $\omega = 1$ corresponds to the original generation. As $\omega$ increases, performance initially improves, reaches an optimum, and subsequently declines with further increases in $\omega$. This simple pattern applies to all the settings. We find $\omega$ is both tunable and transferable. While the optimal value may vary slightly across models and benchmarks, it is generally model-dependent. A value of $\omega = 1.25$ consistently yields robust improvements across all settings. Further details are provided in Appendix J.

To make SPA more stable and eliminate the need of tuning, we also propose a confidence-modulated weighting strategy that adjusts each token's anchoring strength based on its original probability. Details are provided in Appendix M.

### 5.7. Evaluation on Other Generative Tasks

*Table 7.* SPA's performance on different generative tasks ("TRUTH." and "INFO." are evaluation metrics of TruthfulQA).

| MODEL | TRUTH. | INFO. | GSM8K | MMLU | BOOLQ |
|---|---|---|---|---|---|
| LLAMA-3.1 | 88.4 | 97.8 | 76.5 | 69.5 | 83.0 |
| + SPA | +3.1 | +0.9 | +1.1 | +0.0 | +0.1 |

While we focus on code generation in this work, we are interested in whether SPA can be applied to other generative tasks, Thus, we evaluate SPA on other generative benchmarks, including TruthfulQA (Lin et al., 2022), GSM8K (Cobbe et al., 2021), MMLU (Hendrycks et al., 2021b), and BoolQ (Clark et al., 2019). As shown in Table 7,

SPA enhances the base model's performance in all benchmarks except MMLU. While the performance improvement is not as significant as in code generation, we hypothesize that this is because different tasks have unique input and output patterns, leading to varying degrees of attention dilution. For MMLU, the model only needs to generate a choice to answer the multiple-choice question. The output length is significantly shorter than in code generation and considerably less than in user prompts. Consequently, the model's attention is hardly diluted by self-generated tokens, making SPA not helpful. In contrast, TruthfulQA requires the model to generate a text analysis, provide reasoning, and answer the question. Therefore, SPA is more beneficial in addressing attention dilution and correcting the remaining 27% errors for "TRUTH." (Truthfulness) and 41% errors for "INFO." (Informativeness). Nevertheless, it remains an interesting future work to investigate how to further improve SPA for a broader range of generative tasks beyond code generation. More details are reported in Appendix I.

## 6. Related Work

**Code Generation.** To enhance the performance of LLMs on coding tasks, significant efforts have been dedicated to curating high-quality training data (Li et al., 2023; Guo et al., 2024; Tian et al., 2025; Wei et al., 2023) and designing domain-specific training objectives (Niu et al., 2022; Chakraborty et al., 2022). Furthermore, techniques such as instruction tuning (Wei et al., 2022), reinforcement learning with human feedback (Ouyang et al., 2022), and repository-level context modeling (Zhang et al., 2023a) have been explored to improve alignment, reasoning, and context understanding abilities in code generation. However, these approaches often require significant fine-tuning effort. In the meantime, a line of work has focused on developing interactive methods (Di & Zhang, 2025; Tian et al., 2023; 2024) that incorporate real-time human feedback to guide and refine the model's output. Despite their effectiveness, such human-in-the-loop approaches remain limited due to the requirement of user intervention. To overcome these challenges, recent research has increasingly explored automated and training-free prompting methods (Chen et al., 2024; Zhang et al., 2023b; Suzgun et al., 2022; Le et al., 2024), which aim to automatically optimize prompts by integrating additional reasoning or contextual information. For example, Self-Debugging (Chen et al., 2024) enable LLMs to debug code based on error messages and execution results. Self-Planning (Jiang et al., 2024) allows LLMs to decompose tasks into subtasks and implement solutions step-by-step. ReAct (Yao et al., 2023) prompts an LLM to generate reasoning traces and action plans in an interleaved manner. Compared with these methods, SPA takes an orthogonal approach that amplifies the influence of the user prompt to mitigate the attention dilution issue.

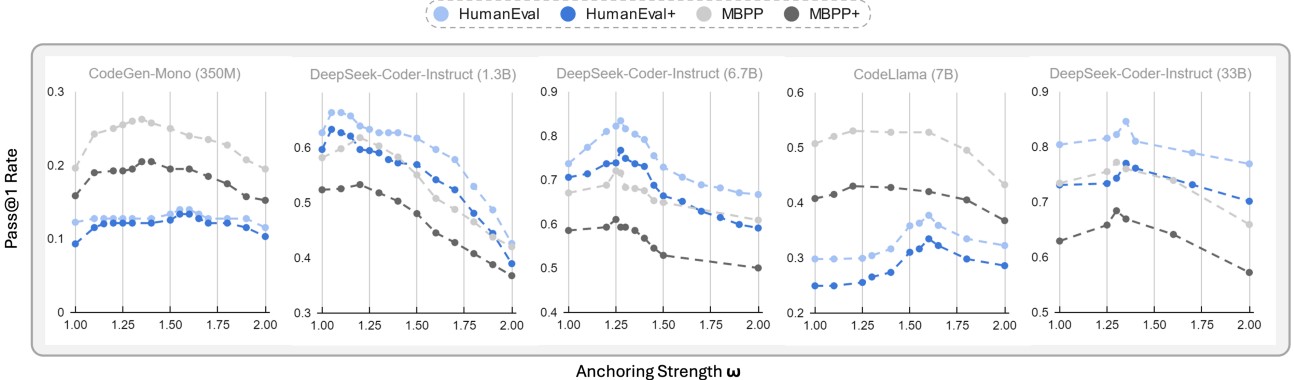

*Figure 3.* Analysis of Anchoring Strength

**Attention Steering.** TOAST (Shi et al., 2023a) tunes a feature selection module to redirect attention to task-relevant features. PASTA (Zhang et al., 2024) performs model profiling to identify beneficial attention headers and recalculate attention distributions across transformer layers. However, these attention-steering approaches usually require extensive model adaptations and complex setup procedures. In contrast, SPA provides a model-agnostic solution that mathematically simulates attention steering via logit arithmetic.

**Logit Arithmetic.** There has been a growing body of research on performing arithmetic transformations on logits to enhance text generation, such as contrasting logits from multiple LMs (Liu et al., 2024a; 2021; Dou et al., 2019; Zhao et al., 2024) and contrasting logits from different layers of a model (Chuang et al., 2024; Gera et al., 2023). Unlike these methods, SPA contrasts logits from the same model by perturbing the input through masking, rather than providing additional context (Pei et al., 2023; Shi et al., 2023b; Malkin et al., 2022) or changing to a completely new prompt (Sennrich et al., 2024). Furthermore, we provide a mathematical approximation of semantic scaling over arbitrary groups of embeddings. SPA is specifically designed to address the attention dilution issue in LLMs during code generation—a phenomenon first observed in our work. By contrast, none of the existing works explored code generation or model attention. They primarily focus on enhancing coherence (Malkin et al., 2022), factuality (Shi et al., 2023b; Chuang et al., 2024; Sennrich et al., 2024; Leng et al., 2023), and controllability (Liu et al., 2021; Pei et al., 2023; Zhao et al., 2024).

## 7. Limitations & Future Work

In this work, we pre-define the method for selecting anchored tokens and use a fixed anchoring strength when generating code. We consider this approach a baseline. Future work could explore how to dynamically select the *anchored text* and *anchoring strength*. For the selection of *anchored text*, one idea is to use LLMs to dynamically identify relevant words or phrases corresponding to the current generation step. For tasks where NL may not be important compared to code (e.g., code translation), we can use static code analysis to identify important code elements (e.g., function calls and variable names heavily used in the code). For the selection of *anchoring strength*, one idea is to develop a method to calculate the relevance of words and phrases to each generation step. Based on the relevance scores, the system can assign higher values to more relevant contexts while assigning lower values to less relevant ones.

In our current experiments, we utilize existing test cases in the benchmark to determine whether to activate SPA. Although this is a common practice in code generation (Chen et al., 2024; Zhang et al., 2023b), this may not be applicable to programming tasks where test cases are not available. This can be potentially addressed by prompting LLMs to generate test cases. Furthermore, we believe SPA will be beneficial in a self-improving pipeline for fairly complex tasks. By analyzing the errors in the initially generated code, LLMs can be prompted to identify which instructions or requirements were not followed in the prompt. Then SPA can be used to amplify the influence of these ignored instructions.

## 8. Conclusion

We present SPA, a model-agnostic approach to enhancing LLM code generation. Our study first identifies the attention dilution phenomenon, where code LLMs increasingly overlook the prompt as generation progresses. To address this, SPA introduces a training-free, mathematically proven mechanism for controlling the influence of selected prompt tokens. Experiments demonstrate that aligning model attention to user prompts using SPA significantly and consistently enhances the code generation performance of base LLMs.

## Acknowledgements

We thank all the anonymous reviewers for their valuable and detailed feedback, which has significantly improved the quality of this paper. This work was supported in part by the National Science Foundation (NSF CAREER Grant 2340408).

## Impact Statement

This paper presents work whose goal is to advance the field of Machine Learning. There are many potential societal consequences of our work, none of which we feel must be specifically highlighted here.

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

## Table of Contents

## A. Performance Analysis by Different Generation Lengths

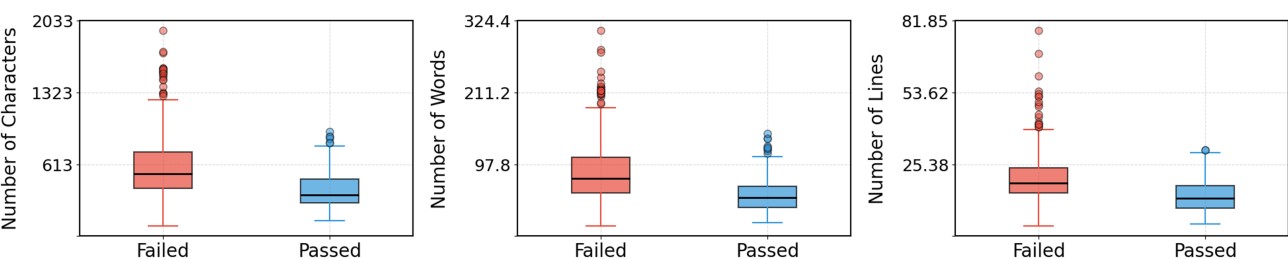

*Figure 4.* Length of correct vs. incorrect code generated by LLMs at three levels of granularity: character, word, and line.

To evaluate how code length affects model performance, we analyze the length distribution of correctly and incorrectly generated code. We use the same set of experimental LLMs (Nijkamp et al., 2023; Guo et al., 2024; Rozière et al., 2024) as in Section 2 and experiment on both HumanEval (Chen et al., 2021) and LiveCodeBench (Jain et al., 2024).

Particularly, we evaluate code length at three granularities: characters, words (tokens), and lines. Figure 4 demonstrates

that incorrectly generated code is significantly longer than correctly generated code across different granularity levels. To mitigate the impact of the correlation between task difficulty and generation length, we conduct further analysis at different task difficulty levels in LiveCodeBench. Table 1 shows that the average length of incorrectly generated code is consistently longer than that of correctly generated code, across all levels.

These results echo our findings about attention dilution—as the generated sequence grows longer, the model's attention to the user prompt diminishes, leading to more errors.

## B. Calculation Details and Extended Discussion of LLM Attention

### B.1. Calculation of Self-attention

Most LLMs are based on the decoder of transformer (Vaswani et al., 2017), which has multiple self-attention layers. Roughly speaking, given an LLM $f_\theta$ and an input sequence of tokens $t_0, t_1, \ldots, t_n$ where $t_i$ represents the $i$th token. The transformer calculates relevance scores between every pair of tokens. The self-attention score for a token $t_i$ in the sequence can be roughly formulated as:

$$\text{attention}(t_i) \approx \frac{\sum_{j=1}^{n} \text{relevance}(t_i, t_j)}{\sum_{i=1}^{n} \sum_{j=1}^{n} \text{relevance}(t_i, t_j)}, \tag{15}$$

where the relevance function approximates the computation among $Q, K, V$ in transformers (Vaswani et al., 2017). However, different layers have different attention distributions. According to a study (Wan et al., 2022b), deeper self-attention layers can better capture long-distance dependencies and program structure, so we calculate the attention by aggregating attention from multiple heads at the last layer. Nevertheless, this still excludes the influence from the last forward layer.

### B.2. Calculation of Gradient-based Attention

To validate the generalizability of attention dilution, we employ a gradient-based attention calculation method (Selvaraju et al., 2016; Shrikumar et al., 2017). Compared to using self-attention layers in transformers, the gradient-based method can be generalized to different model architectures by treating the entire model as a whole differentiable function. It computes the model's attention by calculating the gradients relative to each input token. Intuitively, a token that induces a larger gradient is considered more influential, suggesting that the model pays greater attention to it. Formally, the attention over the token $t_i$ is calculated by

$$\text{attention}(t_i) = \frac{\partial f_\theta(t_0, t_1, \ldots, t_n)}{\partial t_i}. \tag{16}$$

As shown in Figure 5, we observe a similar declining pattern in the model's attention over the initial prompt, suggesting that attention dilution is a fundamental phenomenon that persists across different attention measurement approaches.

### B.3. Attention Misalignment

Despite the success of the attention mechanism in LLMs, prior works found that language models often exhibit simple attention patterns (Raganato & Tiedemann, 2018; Voita et al., 2019). Furthermore, an empirical study (Kou et al., 2024) found that given a coding task, there often exists a misalignment between LLM attention and human attention. When generating code, LLMs often focus on parts in natural language descriptions that are different from what human programmers focus on. Another work (Ning et al., 2024) shows that when the model's attention aligns more closely with human programmers' attention, the model generates more accurate SQL queries. Inspired by these findings, we hypothesize that a root cause of inaccuracy in LLM-generated code stems from the suboptimal model attention.

### B.4. Attention to the User Prompt Ratio

Based on these two methods to calculate LLMs' attention, we analyze how the attention of LLMs to the initial prompt shifts. Formally, given the prompt $x$ and the following generated tokens $t_0, t_1, \ldots, t_{i-1}$, we calculate the *attention to the user prompt ratio* $\alpha(x)$ over the initial prompt

$$\alpha(x) = \frac{\text{attention}(x)}{\text{attention}(x) + \sum_{i=1}^{n} \text{attention}(t_i)} \tag{17}$$

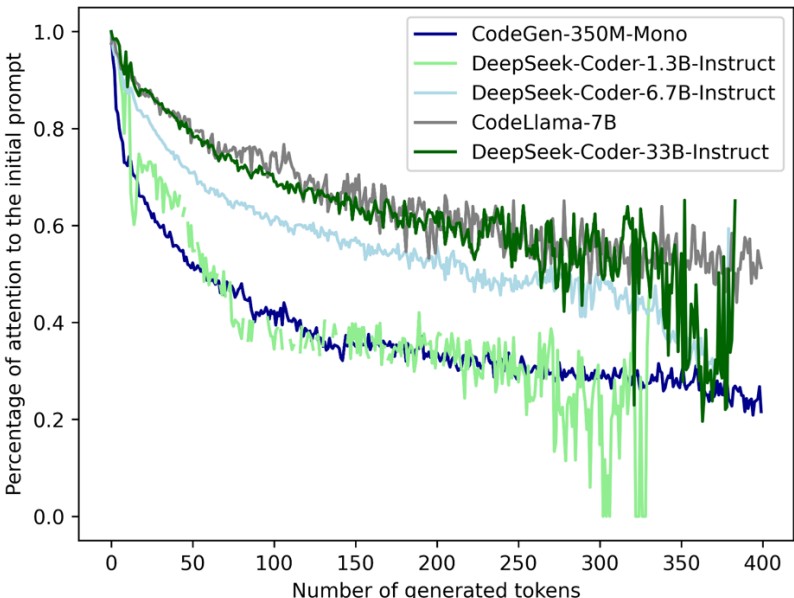

*Figure 5.* Shift of LLMs' gradient-based attention to the initial prompt. The gradient is calculated with respect to the output logits.

Given that attention analysis requires open sourcing, we select five SOTA code LLMs with various sizes. We run the experiments on HumanEval (Chen et al., 2021), one of the most popular benchmarks for evaluating code generation models. We run five LLMs (Nijkamp et al., 2023; Rozière et al., 2024; Guo et al., 2024) on all 164 Humaneval tasks. Figure 5 and Figure 2 show the gradient-based attention shift when generating the first 400 tokens. The value gradually becomes noisy due to the insufficient length of the generated sequence.

The results demonstrate that there indeed exists such an attention dilution issue. Due to the autoregressive nature, LLMs' attention to the initial prompt is gradually diluted as they generate more code. LLMs tend to attend to code generated by themselves. Our finding is supported by another study (Chiang & Cholak, 2022) which investigates the self-attention dilution of transformers in a more general scenario.

## C. Extended Discussion of Approximation in SPA

In Section 3.3, Equation 9 delivers the approximation by only keeping the first derivative in Equation 8, but it is also feasible to calculate a higher-order approximation. For example, if we want to keep the term involving the second-order derivative $\frac{\omega^2}{2!}F_{\theta,i,x}{}''(0)$, it can still be computed using finite-difference methods:

$$F_{\theta,i,x}{}''(0) \approx \frac{F_{\theta,i,x}(1) - 2F_{\theta,i,x}(0) + F_{\theta,i,x}(-1)}{(1-0)^2}. \tag{18}$$

$F_{\theta,i,x}(-1)$ can be solved by Equation 12 where $F_{\theta,i,x}(0)$ and $F_{\theta,i,x}(1)$ are the logits generated from the original input and the logits generated from the masked input.

However, no matter how many terms we keep in Equation 8, we find we can only represent $F_{\theta,i,x}(\omega)$ as a linear combination of $F(0)$ and $F(1)$, weighted by an unknown variable $\omega$.

In Section 5.6, our experiments reveal that $\omega$'s impact on code generation performance follows an unimodal pattern—initially increasing, then decreasing. Due to its distribution simplicity, we argue that while a higher-order approximation may yield a more reasonable performance distribution across different $\omega$ values, it does not significantly affect the process of locating the optimal anchoring strength. Therefore, beyond its computational efficiency, the first-order approximation in SPA is adequate for calculating semantically accurate augmented logits.

# D. Code Generation Examples

Figure 6 presents two examples comparing the code generated by models alone and the models augmented using SPA.

In the first example, CodeLlama (7B) overlooks the specified condition "upper vowels." In contrast, SPA enhances the model's focus on the intended purpose. The code initializes all the upper vowels in the first line and correctly refers to it later.

In the second example, DeepSeek-Coder (1.3B) erroneously sorts the list by string names instead of integers. When using SPA, the model demonstrates improved recognition of the required procedures, aligning more closely with the task description. The code correctly sorts and reverses the list. Then the integer list is mapped to the string list.

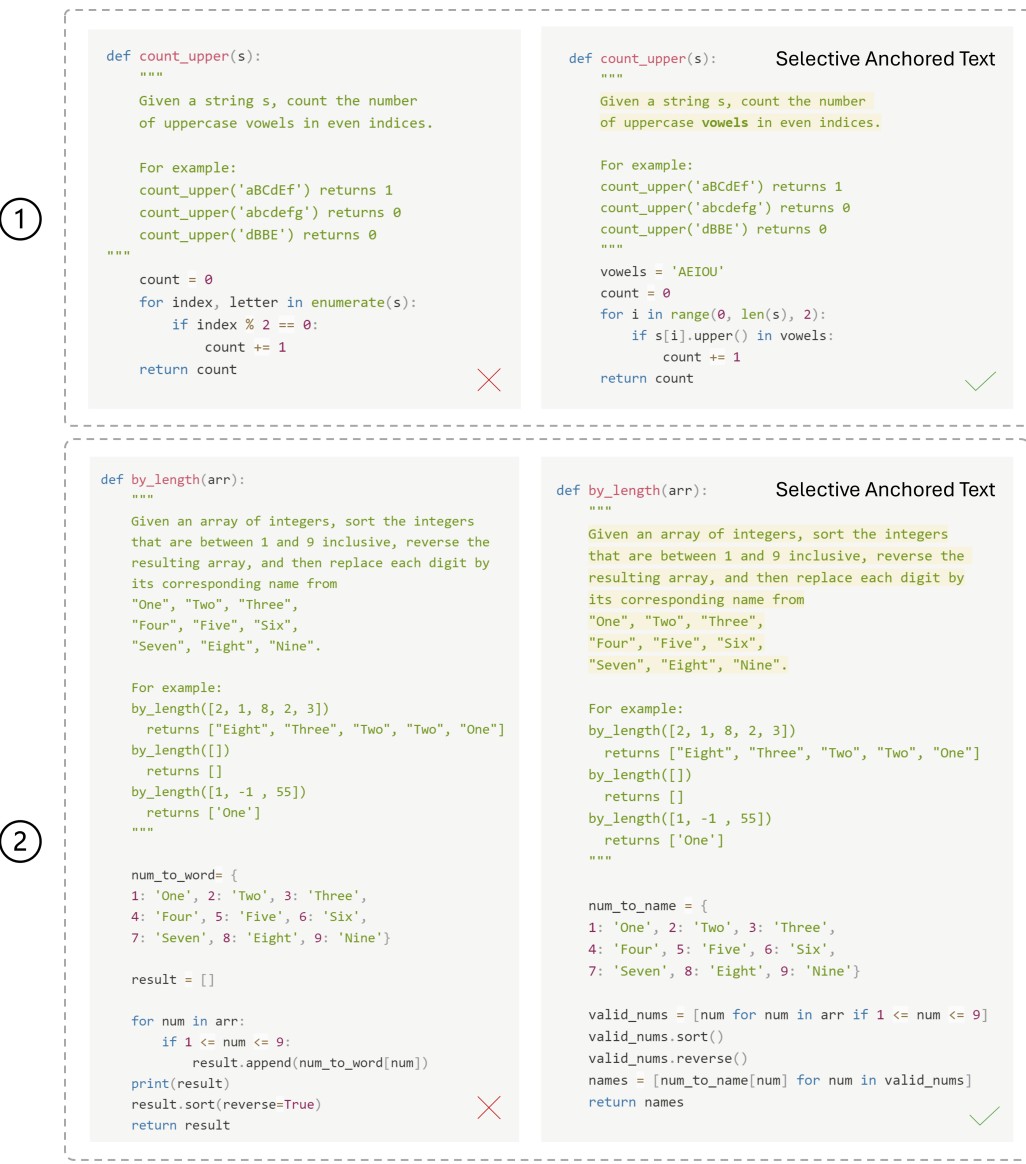

*Figure 6.* Examples of generated code by LLMs alone (left) and using SPA (right).

# E. Computational Cost of SPA

## E.1. Latency

We analyzed the sources of inference overhead, which include running test cases, the two decoding processes, and the logit arithmetic operations. We observed that each attention augmentation in SPA results in an additional 0.6 times the base model's latency, but this increase occurs only when test cases fail. Running test cases incurs only a very small cost (0.1s on average) compared to the total inference time (9.6s). Thus, only activating SPA when test failures are detected can reduce the overall overhead. The more accurate the base model, the less overhead SPA introduces. On average, SPA increases the inference time to 1.27 times that of the original model. We believe this overhead is acceptable in practical development.

*Table 8.* Comparison of Tokens Per Second With and Without SPA.

| MODEL | TOKEN/SECOND |
|---|---|
| CODEGEN (350M) | 34.1 |
| +SPA | 22.7 |
| DEEPSEEK-CODER (1.3B) | 17.8 |
| +SPA | 14.4 |
| DEEPSEEK-CODER (6.7B) | 12.1 |
| +SPA | 10.2 |
| CODELLAMA (7B) | 14.5 |
| +SPA | 10.6 |
| STARCODER (15B) | 7.4 |
| +SPA | 6.2 |
| DEEPSEEK-CODER (33B) | 5.3 |
| +SPA | 4.6 |

## E.2. Memory

SPA requires the storage of the logit generated by the masked prompt embeddings. Theoretically, the additional memory requirement, denoted as $M_{overhead}$, can be expressed as:

$$M_{overhead} = V \times D \times S_{logit} \tag{19}$$

where $V$ represents the vocabulary size, $D$ is the token embedding dimension, and $S_{logit}$ is the size of a single logit value.

Consider a vocabulary size $V = 50,000$, a token embedding dimension $D = 4,096$, and a logit size $S_{logit} = 2$ bytes. Then the additional memory overhead is calculated as:

$$50,000 \times 4,096 \times 2 \text{ bytes} \approx 390 \text{ MB} \tag{20}$$

In practice, this memory overhead can be significantly reduced because many low-ranked tokens hardly contribute to the results. We can calculate the augmented logits by considering only the top-ranked tokens and ignoring the rest. For example, if we focus only on the top 100 logits, the overhead will dramatically decrease to 800 KB.

## E.3. Tradeoff Between Latency and Memory Usage via Parallelization

Currently, SPA is implemented by sequentially computing the logits. However, the latency can be further reduced by parallelizing the logits generation from the original and masked embeddings, as these computations are independent (Figure 1). This optimization, however, comes at the cost of increased memory usage (double VRAM or memory for forwarding). Given that LLM inference is memory-intensive, this introduces a tradeoff between reduced latency and higher memory consumption.

# F. Pass@10 & Beam Search with SPA

To further evaluate the generalizability of SPA, we assess its Pass@10 performance using beam search. While SPA produces augmented logits that could be used directly for beam search, we observed that directly sampling top beams from these

augmented logits does not improve the performance. We hypothesize that this phenomenon occurs because while SPA successfully amplifies the influence of anchored text and improves the accuracy of top logits, it also amplifies noise in lower-ranked logits. This undermines the reliability of the overall probability distribution, thereby hindering the sampling process.

To address this issue, we retrieve top candidate tokens based on the augmented logits but use original probabilities to accumulate beam probability. This strategy ensures that important, potentially overlooked tokens are considered while maintaining reliable probabilities.

*Table 9.* Pass@1 and Pass@10 (%) with and without using SPA.

| MODEL | SIZE | HUMANEVAL | | HUMANEVAL+ | | MBPP | | MBPP+ | |
|---|---|---|---|---|---|---|---|---|---|
| | | PASS@1 | PASS@10 | PASS@1 | PASS@10 | PASS@1 | PASS@10 | PASS@1 | PASS@10 |
| CODEGEN-MONO | (350M) | 15.3 | 36.6 | 12.2 | 33.6 | 19.6 | 47.7 | 15.9 | 42.4 |
| + SPA | | 20.1 (+4.8) | 39.0 (+2.4) | 17.1 (+4.9) | 38.5 (+4.9) | 27.4 (+7.8) | 55.6 (+7.9) | 22.6 (+6.7) | 43.3 (+0.9) |
| DEEPSEEK-CODER | (1.3B) | 66.4 | 73.3 | 61.8 | 68.7 | 58.2 | 67.0 | 52.4 | 63.7 |
| + SPA | | 70.1 (+3.7) | 73.3 (+0.0) | 67.7 (+5.9) | 69.3 (+0.6) | 60.9 (+2.7) | 68.8 (+1.8) | 52.4 (+0.0) | 64.3 (+0.6) |
| DEEPSEEK-CODER | (6.7B) | 75.6 | 84.0 | 70.2 | 77.9 | 67.0 | 79.8 | 58.5 | 70.2 |
| + SPA | | 88.5 (+12.9) | 86.4 (+2.4) | 79.9 (+9.7) | 82.8 (+4.9) | 71.0 (+4.0) | 87.7 (+7.9) | 60.7 (+2.2) | 73.9 (+3.7) |
| CODELLAMA | (7B) | 33.6 | 58.0 | 28.2 | 48.9 | 50.9 | 61.0 | 40.8 | 49.0 |
| + SPA | | 44.0 (+10.4) | 64.7 (+6.7) | 36.0 (+7.8) | 54.4 (+5.5) | 54.3 (+3.4) | 65.3 (+4.3) | 44.0 (+3.2) | 52.0 (+3.0) |
| DEEPSEEK-CODER | (33B) | 81.7 | 88.5 | 77.1 | 80.2 | 73.4 | 86.8 | 63.2 | 75.8 |
| + SPA | | 86.2 (+4.5) | 89.7 (+1.2) | 79.3 (+2.2) | 83.2 (+3.0) | 79.4 (+6.0) | 89.2 (+2.4) | 70.3 (+7.1) | 80.1 (+4.3) |

As shown in Table 9, SPA consistently improves Pass@10 by, on average, 3.42% and up to 7.9% when using beam search. While the improvements are not as pronounced as those seen with Pass@1, we anticipate that future work could develop beam search algorithms specifically optimized for SPA's unique logit distribution characteristics.

## G. Hypothetical explanation for Attention dilution and SPA's effectiveness

SPA is motivated by a recent study (Kou et al., 2024) and our empirical observations demonstrating the attention dilution issue. Our experiment results in Section 5 echo our observation and confirm the existence of attention dilution during code generation. Here we propose a detailed explanation for this phenomenon based on our knowledge and hypotheses. We believe it stems from two limitations in regular autoregressive decoding: (1) **Distraction** and (2) **Error propagation**.

**Distraction.** When a transformer generates a token, its correctness depends on two abilities: (1) whether the model attends to the correct context, and (2) whether the model can derive the correct token based on this context. SPA aims to improve the first ability. Suppose we have a perfect transformer. For each generated token, it should only attend to relevant prior tokens and ignore irrelevant ones. However, no model is perfect. For each prior token, there is a chance the model incorrectly identifies and attends to it. As the model generates more tokens that compete for attention, it becomes increasingly challenging for the model to accurately distribute its attention. The model has more chances to attend to irrelevant tokens, making its attention increasingly unreliable.

In contrast, the user prompt is persistently relevant throughout the generation since it represents the user's intent. While self-generated tokens are also important context, they are less persistently related than task descriptions in code generation. Amplifying the impact of the task description by SPA essentially enhances attention reliability, thereby mitigating distraction.

**Error propagation.** During code generation, the model may generate irrelevant code tokens. However, the autoregressive nature of LLMs assumes that all previously generated tokens are correct. For example, if the model introduces an irrelevant variable declaration, subsequent generations may take it into account and continue to generate irrelevant code. Although the model can still generate correct code behavior in later generations, this assumption of correctness makes it difficult to identify errors. As a result, the error can propagate and accumulate, leading to a higher probability of errors in later generations.

SPA mitigates this issue by reinforcing attention to the user prompt while downplaying reliance on self-generated tokens.

This optimizes the attention distribution based on the trustworthiness of different contexts, thereby increasing accuracy.

## H. Implementation and Deployment

### H.1. Implementation of SPA

SPA is a model-agnostic algorithm and our implementation does not rely on specific models. All six models in our paper are built upon the Huggingface Transformer library, which offers APIs to directly access and edit token embeddings and logits. Particularly, the SPA generator inherits the Huggingface Transformers generation API.[9] We leverage the hook to modify the logit calculation within the original generation pipeline. The API works for any LLM in the Huggingface model collections [10] with native hyperparameters such as TEMPERATURE. We have released a PyPI library [11] for developers to quickly test SPA.

### H.2. Model Deployment

We downloaded and deployed LLMs from Huggingface. To expedite evaluations, we apply 8-bit quantization (Frantar et al., 2023; Dettmers et al., 2022) to all models. Prior studies (Li et al., 2024; Huang et al., 2024) have demonstrated that this approach has very little impact on LLM performance. We set the *Temperature* to 0 and the *beam* to 1 for greedy decoding in all experiments, except for the one described in Appendix F. All experiments were conducted on a 64-bit Ubuntu 22.04 LTS system, equipped with an AMD EPYC 7313 CPU, eight NVIDIA A5500 GPUs, and 512GB of memory. The experiments ran for approximately seven weeks.

### H.3. Prompt Design

We use the original task descriptions from the datasets as prompts for the text-completion models, CodeLlama and CodeGen-Mono. For the remaining models, we format the prompts using the official chat template from HuggingFace. All experiments are conducted in a zero-shot setting.

## I. Evaluating SPA on Other Generative Tasks

To evaluate the generalizability of SPA beyond code generation, we experiment SPA on other generative tasks, including TruthfulQA (Lin et al., 2022), GSM8K (Cobbe et al., 2021), MMLU (Hendrycks et al., 2021b), and BoolQ (Clark et al., 2019).

TruthfulQA (Lin et al., 2022) is a benchmark designed to measure models' ability to avoid generating false or misleading information, requiring models to answer questions while remaining truthful. GSM8K (Cobbe et al., 2021) tests mathematical reasoning capabilities through grade school math word problems that require multi-step solutions. MMLU (Hendrycks et al., 2021b) evaluates models across 57 subjects, including elementary mathematics, US history, computer science, law, and more, comprehensively testing both breadth and depth of knowledge. It provides multiple-choice questions for LLMs to identify the correct answers. BoolQ (Clark et al., 2019) consists of naturally occurring yes/no questions from web queries, testing reading comprehension and binary classification abilities. We use Llama 3.1-Instruct-8B as our base model since the other models in our study are specifically fine-tuned for code tasks.

*Table 10.* Evaluating SPA on Different Generative Tasks.

| MODEL | TRUTHFULQA (TRUTH.) | TRUTHFULQA (INFO.) | GSM8K | MMLU | BOOLQ | HUMANEVAL |
|---|---|---|---|---|---|---|
| LLAMA-3.1-INSTRUCT-8B | 88.40% | 97.78% | 76.5% | 69.5% | 83% | 63.4% |
| + SPA | + 3.14% | + 0.91% | + 1.10% | + 0% | + 0.03% | **+ 9.76%** |

As shown in Table 7, while SPA provides improvements across these tasks, the gains are significantly smaller compared to

---

[9] https://huggingface.co/docs/transformers/en/main_classes/text_generation
[10] https://huggingface.co/models
[11] https://pypi.org/project/anchoring/

code generation tasks. We attribute this performance difference to the unique input-output pattern of different generative tasks.

For MMLU, the model only needs to generate a choice to answer the multiple-choice question. The output length is significantly shorter than in code generation and considerably less than in user prompts. Consequently, the model's attention is hardly diluted by self-generated tokens, making SPA not helpful. In contrast, TruthfulQA requires the model to generate a text analysis, provide reasoning, and answer the question. Therefore, SPA is more beneficial in addressing attention dilution and correcting the remaining 27% errors for Truthfulness and 41% errors for Informativeness. In contrast, code generation is typically lengthy and can easily lead to attention dilution. Moreover, code generation prompts serve as persistent instructions, requiring the LLM to maintain focus throughout the generation process. This differs from tasks like translation, where there is no inherent need to consistently anchor attention to specific components of the input.

Therefore, we believe SPA is especially suitable for enhancing LLMs' attention for code generation tasks. Nevertheless, it remains an interesting future work to investigate how to further improve SPA for a broader range of generative tasks beyond code generation. More details are reported in Appendix I.

## J. Tuning of Anchoring Strength

As demonstrated in Section 5.6, different anchoring strengths $\omega$ lead to different performance and follow a simple unimodal pattern. In this section, we report more evaluation of tuning SPA. To ensure fair comparisons, SPA is activated for all tasks in all experiments, unlike during inference time where SPA is only activated for failed tasks. Moreover, we tune SPA on all the tasks for better generalizability.

### J.1. Tuning Stability

We evaluate the stability of tuning anchoring strength by tuning SPA on five exclusive subsets of each dataset for each model, as shown in Table 11. The average variance across subsets is 0.0046 for HumanEval/HumanEval+ and 0.0026 for MBPP/MBPP+, demonstrating the tuning stability.

### J.2. Cross-dataset & Cross-model Tuning Evaluation

We investigate the transferability of this hyperparameter across different models and datasets. Firstly, we conduct a *cross-dataset* evaluation between HumanEval/HumanEval+ and MBPP/MBPP+, which have distinct prompt formats. We tune $\omega$ on HumanEval+ and evaluate Pass@1 on MBPP and MBPP+, and vice versa[12] (denoted as SPA$_{cross-dataset}$). We calculate average Pass@1 improvements on the original and plus versions across all baseline models. Secondly, we perform a *cross-model* evaluation by tuning $\omega$ on one model and evaluating Pass@1 on the remaining four. For each model, we compute the average Pass@1 improvements across all the other models, for HumanEval/HumanEval+ and MBPP/MBPP+ respectively (denoted as SPA$_{cross-model}$). Similar to Section 5, SPA represents tuning within the split partial dataset, while SPA* represents tuning within the entire dataset.

As shown in Table 12, we find the anchoring strength $\omega$ tuned on one model is hardly transferred to another. However, $\omega$ tuned on one dataset can be transferred to another with reduced but still effective performance. These observations suggest that the anchoring strength is highly model-dependent and partially task-dependent.

We further investigate whether it is possible to find a universal anchoring strength that works for most scenarios. One potential value is the average of the tuned anchoring strengths across benchmarks for each model. We apply this fixed anchoring strength to all settings, denoted as SPA-preset. As shown in Table 13, although the generation accuracy decreases compared to using a tuned strength, SPA with the fixed strength can still outperform the baselines. It implies that SPA is effectively deployable in new scenarios once a reasonable value is set.

### J.3. Comparison between Tuned SPA and Optimal SPA

To better understand the tuning efficiency, we tune SPA on each complete dataset to obtain the optimal anchoring strength (denoted as SPA*). As shown in Table 14, we find that SPA* achieves 20% higher performance improvements compared to

---

[12] The "plus" versions of HumanEval and MBPP share identical prompts with their base counterparts, so we only tune once on the plus version.

*Table 11.* Tuned Anchoring Strength on Different Subsets.

| MODEL | SUBSET | HUMANEVAL/+ | MBPP/+ |
|---|---|---|---|
| CODEGEN-MONO (350M) | SUBSET$_1$ | 1.05 | 1.30 |
| | SUBSET$_2$ | 1.10 | 1.35 |
| | SUBSET$_3$ | 1.20 | 1.25 |
| | SUBSET$_4$ | 1.30 | 1.35 |
| | SUBSET$_5$ | 1.25 | 1.35 |
| | COMPLETE | 1.20 | 1.35 |
| DEEPSEEK-CODER (1.3B) | SUBSET$_1$ | 1.05 | 1.20 |
| | SUBSET$_2$ | 1.05 | 1.15 |
| | SUBSET$_3$ | 1.10 | 1.15 |
| | SUBSET$_4$ | 1.00 | 1.20 |
| | SUBSET$_5$ | 1.05 | 1.25 |
| | COMPLETE | 1.05 | 1.20 |
| DEEPSEEK-CODER (6.7B) | SUBSET$_1$ | 1.30 | 1.30 |
| | SUBSET$_2$ | 1.25 | 1.20 |
| | SUBSET$_3$ | 1.30 | 1.25 |
| | SUBSET$_4$ | 1.20 | 1.20 |
| | SUBSET$_5$ | 1.35 | 1.25 |
| | COMPLETE | 1.28 | 1.25 |
| CODELLAMA (7B) | SUBSET$_1$ | 1.55 | 1.25 |
| | SUBSET$_2$ | 1.55 | 1.20 |
| | SUBSET$_3$ | 1.50 | 1.20 |
| | SUBSET$_4$ | 1.65 | 1.25 |
| | SUBSET$_5$ | 1.65 | 1.15 |
| | COMPLETE | 1.60 | 1.20 |
| DEEPSEEK-CODER (33B) | SUBSET$_1$ | 1.25 | 1.25 |
| | SUBSET$_2$ | 1.30 | 1.30 |
| | SUBSET$_3$ | 1.40 | 1.30 |
| | SUBSET$_4$ | 1.35 | 1.40 |
| | SUBSET$_5$ | 1.35 | 1.20 |
| | COMPLETE | 1.35 | 1.30 |

*Table 12.* Pass@1 improvements (%) based on cross-dataset tuning.

| DATASET | SPA$_{cross-dataset}$ | SPA$_{cross-model}$ | SPA | SPA$^*$ |
|---|---|---|---|---|
| HUMANEVAL/+ | + 2.01 | - 0.29 | + 4.36 | + 5.11 |
| MBPP/+ | + 2.50 | + 0.37 | + 2.86 | + 3.57 |

*Table 13.* Comparison between SPA and SOTA methods.

| METHOD | ΔPASS@1 (%) | TIME (SEC) |
|---|---|---|
| BASE MODEL | 0 | 7.7 |
| PASTA | +1.2 | 48.8 |
| SELF-DEBUGGING | +4.2 | 27.3 |
| SELF-EDIT | +1.8 | 26.4 |
| SELF-PLANNING | +3.6 | 21.6 |
| REACT | +1.3 | 28.8 |
| SPA-FIXED-STRENGTH | +5.0 | **9.8** |
| SPA | **+7.7** | **9.8** |

the tuned SPA on average.

Table 15 reports optimal anchoring strength values $\omega$. We observe that the average value of 1.28 can be used to effectively improve performance across all benchmarks for all LLMs.

*Table 14.* Pass@1 Improvements (%) Comparision between Tuned SPA and Optimal SPA (SPA is applied to all tasks).

| MODEL | SIZE | HUMANEVAL | HUMANEVAL+ | MBPP | MBPP+ |
|---|---|---|---|---|---|
| CODEGEN-MONO | (350M) | 15.3 | 12.2 | 19.6 | 15.9 |
| + SPA | | 18.3 (+3.0) | 16.0 (+3.8) | 24.9 (+5.3) | 20.6 (+4.7) |
| + SPA* | | 18.3 (+3.0) | 16.0 (+3.8) | 24.9 (+5.3) | 20.6 (+4.7) |
| DEEPSEEK-CODER | (1.3B) | 66.4 | 61.8 | 58.2 | 52.4 |
| + SPA | | 69.5 (+3.1) | 66.4 (+4.6) | 59.1 (+0.9) | 52.4 (+0.0) |
| + SPA* | | 71.0 (+4.6) | 66.4 (+4.6) | 61.7 (+3.5) | 53.4 (+1.0) |
| DEEPSEEK-CODER | (6.7B) | 75.6 | 70.2 | 67.0 | 58.5 |
| + SPA | | 83.2 (+7.6) | 75.6 (+5.4) | 69.6 (+2.6) | 60.2 (+1.7) |
| + SPA* | | 84.0 (+8.4) | 76.3 (+6.1) | 72.2 (+5.2) | 61.1 (+2.6) |
| CODELLAMA | (7B) | 33.6 | 28.2 | 50.9 | 40.8 |
| + SPA | | 40.5 (+6.9) | 33.6 (+5.4) | 52.9 (+2.0) | 43.1 (+2.3) |
| + SPA* | | 41.2 (+7.6) | 35.9 (+7.7) | 52.9 (+2.0) | 43.1 (+2.3) |
| DEEPSEEK-CODER | (33B) | 81.7 | 77.1 | 73.4 | 63.2 |
| + SPA | | 84.7 (+3.0) | 77.9 (+0.8) | 77.2 (+3.8) | 68.5 (+5.3) |
| + SPA* | | 85.5 (+3.8) | 78.6 (+1.5) | 77.2 (+3.8) | 68.5 (+5.3) |

*Table 15.* Optimal Anchoring Strength ($\omega$) for each model and benchmark.

| MODEL | HUMANEVAL | HUMANEVAL+ | MBPP | MBPP+ | *Average* |
|---|---|---|---|---|---|
| CODEGEN-MONO (350M) | 1.20 | 1.20 | 1.35 | 1.35 | 1.28 |
| DEEPSEEK-CODER (1.3B) | 1.05 | 1.05 | 1.20 | 1.20 | 1.13 |
| DEEPSEEK-CODER (6.7B) | 1.28 | 1.28 | 1.25 | 1.25 | 1.26 |
| CODELLAMA (7B) | 1.60 | 1.60 | 1.20 | 1.20 | 1.40 |
| DEEPSEEK-CODER (33B) | 1.35 | 1.35 | 1.30 | 1.30 | 1.33 |
| *Average* | 1.30 | 1.30 | 1.33 | 1.33 | 1.28 |

# K. Analysis of SPA's Performance without Test Case Availability

To analyze SPA's effectiveness in scenarios without available test cases, we conduct additional experiments where SPA is applied to all code generation tasks (SPA$^{all}$). As shown in Table 17, SPA$^{all}$ can still achieve significant improvements across all benchmarks and models. For example, on HumanEval, SPA$^{all}$ improves DeepSeek-Coder (6.7B)'s Pass@1 by 7.6% (from 75.6% to 83.2%).

*Table 16.* Comparison to PASTA in terms of Pass@1.

| | HUMANEVAL | HUMANEVAL+ | MBPP | MBPP+ | BIGCODEBENCH |
|---|---|---|---|---|---|
| PASTA | +1.22 | +1.22 | +1.17 | +0.94 | +0.1 |
| SPA | +7.32 | +4.88 | +4.22 | +3.51 | +0.4 |

# L. Comparison to PASTA

Unlike mainstream methods that optimize prompts, SPA enhances performance by optimizing model attention. We compare SPA with another attention steering method, PASTA (Zhang et al., 2024). PASTA's implementation only supports LLAMA (Touvron et al., 2023a), LLAMA2 (Touvron et al., 2023b), and GPT-J (Wang & Komatsuzaki, 2021). We find that adapting PASTA to new models requires significant effort, so we only evaluate it on LLAMA-7B, LLAMA2-7B, and GPT-J-6B.

*Table 17.* Absolute ($\Delta$) and Relative ($\uparrow$) Performance improvements in Pass@1 rates (%). SPA$^{all}$ is applied to all tasks, while SPA indicates SPA is activated when the original generated code fails test cases.

| MODEL | SIZE | HUMANEVAL | HUMANEVAL+ | MBPP | MBPP+ | BIGCODEBENCH |
|---|---|---|---|---|---|---|
| CODEGEN-MONO | (350M) | 15.3 | 12.2 | 19.6 | 15.9 | 1.1 |
| + SPA$^{all}$ | | 18.3 $^{\Delta+3.0}_{(20\%\ \uparrow)}$ | 16.0 $^{\Delta+3.8}_{(31\%\ \uparrow)}$ | 24.9 $^{\Delta+5.3}_{(27\%\ \uparrow)}$ | 20.6 $^{\Delta+4.7}_{(30\%\ \uparrow)}$ | 1.4 $^{\Delta+0.3}_{(27\%\ \uparrow)}$ |
| + SPA | | 20.1 $^{\Delta+4.8}_{(31\%\ \uparrow)}$ | 17.1 $^{\Delta+4.9}_{(40\%\ \uparrow)}$ | 27.4 $^{\Delta+7.8}_{(40\%\ \uparrow)}$ | 22.6 $^{\Delta+6.7}_{(42\%\ \uparrow)}$ | 1.6 $^{\Delta+0.5}_{(45\%\ \uparrow)}$ |
| DEEPSEEK-CODER | (1.3B) | 66.4 | 61.8 | 58.2 | 52.4 | 2.5 |
| + SPA$^{all}$ | | 69.5 $^{\Delta+3.1}_{(5\%\ \uparrow)}$ | 66.4 $^{\Delta+4.6}_{(7\%\ \uparrow)}$ | 59.1 $^{\Delta+0.9}_{(2\%\ \uparrow)}$ | 52.4 $^{\Delta+0.0}_{(0\%\ \uparrow)}$ | 3.3 $^{\Delta+0.8}_{(32\%\ \uparrow)}$ |
| + SPA | | 70.1 $^{\Delta+3.7}_{(6\%\ \uparrow)}$ | 67.7 $^{\Delta+5.9}_{(10\%\ \uparrow)}$ | 60.9 $^{\Delta+2.7}_{(5\%\ \uparrow)}$ | 52.4 $^{\Delta+0.0}_{(0\%\ \uparrow)}$ | 3.4 $^{\Delta+0.9}_{(36\%\ \uparrow)}$ |
| DEEPSEEK-CODER | (6.7B) | 75.6 | 70.2 | 67.0 | 58.5 | 12.7 |
| + SPA$^{all}$ | | 83.2 $^{\Delta+7.6}_{(10\%\ \uparrow)}$ | 75.6 $^{\Delta+5.4}_{(8\%\ \uparrow)}$ | 69.6 $^{\Delta+2.6}_{(4\%\ \uparrow)}$ | 60.2 $^{\Delta+1.7}_{(3\%\ \uparrow)}$ | 14.2 $^{\Delta+1.5}_{(12\%\ \uparrow)}$ |
| + SPA | | 88.5 $^{\Delta+12.9}_{(17\%\ \uparrow)}$ | 79.9 $^{\Delta+9.7}_{(14\%\ \uparrow)}$ | 71.0 $^{\Delta+4.0}_{(6\%\ \uparrow)}$ | 60.7 $^{\Delta+2.2}_{(4\%\ \uparrow)}$ | 16.4 $^{\Delta+3.7}_{(29\%\ \uparrow)}$ |
| CODELLAMA | (7B) | 33.6 | 28.2 | 50.9 | 40.8 | 3.4 |
| + SPA$^{all}$ | | 40.5 $^{\Delta+6.9}_{(21\%\ \uparrow)}$ | 33.6 $^{\Delta+5.4}_{(19\%\ \uparrow)}$ | 52.9 $^{\Delta+2.0}_{(4\%\ \uparrow)}$ | 43.1 $^{\Delta+2.3}_{(6\%\ \uparrow)}$ | 3.8 $^{\Delta+0.4}_{(12\%\ \uparrow)}$ |
| + SPA | | 44.0 $^{\Delta+10.4}_{(31\%\ \uparrow)}$ | 36.0 $^{\Delta+7.8}_{(28\%\ \uparrow)}$ | 54.3 $^{\Delta+3.4}_{(7\%\ \uparrow)}$ | 44.0 $^{\Delta+3.2}_{(8\%\ \uparrow)}$ | 4.1 $^{\Delta+0.7}_{(21\%\ \uparrow)}$ |
| STARCODER2 | (16B) | 67.7 | 60.4 | 78.0 | 65.1 | 13.3 |
| + SPA$^{all}$ | | 72.1 $^{\Delta+4.4}_{(6\%\ \uparrow)}$ | 63.6 $^{\Delta+3.2}_{(5\%\ \uparrow)}$ | 80.9 $^{\Delta+2.9}_{(4\%\ \uparrow)}$ | 67.6 $^{\Delta+2.5}_{(4\%\ \uparrow)}$ | 14.1 $^{\Delta+0.8}_{(6\%\ \uparrow)}$ |
| + SPA | | 75.6 $^{\Delta+7.9}_{(12\%\ \uparrow)}$ | 65.6 $^{\Delta+5.2}_{(9\%\ \uparrow)}$ | 82.0 $^{\Delta+4.0}_{(5\%\ \uparrow)}$ | 69.1 $^{\Delta+4.0}_{(6\%\ \uparrow)}$ | 14.3 $^{\Delta+1.0}_{(8\%\ \uparrow)}$ |
| DEEPSEEK-CODER | (33B) | 81.7 | 77.1 | 73.4 | 63.2 | 18.9 |
| + SPA$^{all}$ | | 84.7 $^{\Delta+3.0}_{(4\%\ \uparrow)}$ | 77.9 $^{\Delta+0.8}_{(1\%\ \uparrow)}$ | 77.2 $^{\Delta+3.8}_{(5\%\ \uparrow)}$ | 68.5 $^{\Delta+5.3}_{(8\%\ \uparrow)}$ | 20.7 $^{\Delta+1.8}_{(10\%\ \uparrow)}$ |
| + SPA | | 86.2 $^{\Delta+4.5}_{(6\%\ \uparrow)}$ | 79.3 $^{\Delta+2.2}_{(3\%\ \uparrow)}$ | 79.4 $^{\Delta+6.0}_{(8\%\ \uparrow)}$ | 70.3 $^{\Delta+7.1}_{(11\%\ \uparrow)}$ | 21.5 $^{\Delta+2.6}_{(14\%\ \uparrow)}$ |

Table 16 demonstrates that SPA consistently outperforms PASTA on five benchmarks, achieving 4X higher Pass@1 on average. We attribute this to two hypothetical reasons. First, internally editing model attention can be sensitive and exhibit unexpected behaviors. Second, identifying effective attention headers may not be stable or generalizable across different tasks. In contrast, SPA augments the final logits without internally editing the model's feed-forward process, making it model-agnostic. Furthermore, SPA only introduces a single hyperparameter, and the tuning is stable (discussed in Appendix J.1).

## M. Confidence-Modulated Anchoring Strength

To make SPA more stable and eliminate the need for tuning, we propose a confidence-modulated weighting method that adjusts each token's anchoring strength based on its original probability respectively.

When the LLM is highly confident about a specific token, the corresponding logit is adjusted less, resulting in a lower anchoring weight applied to the logit difference. Conversely, if the LLM shows lower confidence, a relatively higher weight is applied, leading to a greater adjustment of the logit value. The confidence level is measured using the original probability distribution generated by the LLM.

Formally, let $p_t$ represent the predicted probability of token $t$ within the original probability distribution for the next token. The anchoring strength $\omega_t$ for token $t$ is defined as:

$$\omega_t = \lambda \cdot (1 - p_t), \tag{21}$$

where $\lambda$ is a fixed coefficient that controls the overall anchoring strength. This results in a token-wise anchoring strength vector $\omega \in \mathbb{R}^{|\mathcal{V}|}$ over the vocabulary $\mathcal{V}$. The vector $\omega$ is then element-wise multiplied with the original logit difference vector to obtain a reweighted logit difference (Figure 1 ④).

