# OpenReview forum: "Selective Prompt Anchoring for Code Generation"
_ICML.cc/2025/Conference — ICML 2025 poster_

### Official Review · Reviewer_fXfw · 2025-03-07

**Overall Recommendation:** 4

**Summary:**

This paper proposes a method for improving LLM's performance at test time. The method helps the LLM keep the focus on the task, avoiding dilution as the number of tokens generated grows for complex tasks such as coding. The method anchors a part of the prompt that specifies the task to accomplish by amplifying its corresponding logits before next token generation. The method compares the logits generated by the LLM with the orginal token embedding matrix and those generated by an embedding matrix where the part corresponding to the anchor has been masked; the difference in logit strength is used to amplify the output logits. The anchor part of the prompt is defined by special tokens. The empirical evaluation of the method comprises 5 code benchmarks (HumanEval, BigCodeBench, etc.), 6 LLM open-source base models (DeepSeek, CodeGEN, etc.) and 5 SOTA methods (ReACT, PASTA, etc.). Results shows consistent improvements (13%) across all benchmarks compared to the baseline and better performance than SOTA methods. Several ablation and parameter studies are performed to determine the best part of the prompt to mask, the influence of the prompt length, the effect of using other programming languages and the tuning of the anchoring strength.

## update after rebuttal:
After carefully reviewing the author's rebuttal and considering the other reviews, i raise my score to Accept.

**Claims And Evidence:**

Yes, the effects of anchoring part of the prompt to avoid dilution are shown clearly on several empirical studies.

**Essential References Not Discussed:**

I'm not aware of essential references not discussed in the paper.

**Experimental Designs Or Analyses:**

The experimental design is fine, including the evaluation metrics (pass@k). The choice of 6 open-source code LLMs is extensive and representative: CodeGen-Mono-350M (Nijkamp et al.,2023), CodeLlama-7B (Rozi`ere et al., 2024), StarCoder2-15B (Lozhkov et al., 2024), and DeepSeek-Coder-Instruct-1.3B, 6.7B, and 33B (Guo et al., 2024). The choice of SOTA methods to compare to is fine (although i may be unaware of some recent proposed methods).

**Methods And Evaluation Criteria:**

Yes, the evaluation methods are well suited for the application. The code-generation benchmarks used are well-known and effective.

**Other Comments Or Suggestions:**

see above.

**Other Strengths And Weaknesses:**

Strengths:
- the method is general, model agnostic and can be applied beyond coding tasks. The intuition behind the approach is sound and well explained.
- the method is fast as it does only adds 60% over the base model and does not used much extra memory. It is significantly faster than the SOTA methods compared against (> 3x).
- The paper is well written and organized. It is easy to follow.
- Code is provided on anonymous github

Weaknesses:
- The method applies masking to the token embeddings and arithmetic to the logits, so it needs to be inserted within the LLM model. No discussion is made on how complex that is. The modifications would also be model-dependent. It is not discussed how hard it is to implement these modifications for the chosen models. Are there commonalities that can be abstracted ? Overall, this approach is an integral part of an open-source model code, and cannot just be applied on top of an existing models in an easy way.
- The hyperparameter that controls the strength of the logit amplification is model and dataset -dependent. It needs to be tuned as it exhibits a clear maxima. This effect is well discussed in appendix J and the authors show that, while the parameter is strongly model-dependent, the method can still be effective with a fixed value across the benchmark datasets. Unfortunately, the authors do not compare directly in a table the fixed approach with the other SOTA methods. This would be desired, as the real question is whether the proposed approach beats SOTA for a deployable model (dataset-independent).

**Questions For Authors:**

no questions.

**Relation To Broader Scientific Literature:**

The works on code-generation, attention and logit arithmetic are well cited.

**Theoretical Claims:**

The mathematics of the method are clearly explained, including finite-difference approximation used to reduce computational complexity.  There are no theoretical proofs for bounds though.

---

> ### Author Rebuttal · Authors · 2025-04-01
>
> `The method applies masking to the token embeddings and arithmetic to the logits, so it needs to be inserted within the LLM model. No discussion is made on how complex that is. The modifications would also be model-dependent. It is not discussed how hard it is to implement these modifications for the chosen models. Are there commonalities that can be abstracted ? Overall, this approach is an integral part of an open-source model code, and cannot just be applied on top of an existing models in an easy way.`
>
> The implementation is not model-dependent. All six models in our paper are built upon the Huggingface Transformer library, which offers the APIs to directly access and edit token embeddings and logits. So we only need to implement one single decoding method for all these models using these APIs based on the SPA algorithm. Please check Lines 2859-3019 for the implementation of this decoding method in our anonymous repository (https://anonymous.4open.science/r/Selective-Prompt-Anchoring-3693/weighted_utils/weighted_text_utils.py). We currently cannot integrate SPA with closed-source models like GPT-4 since these models only provide a text-based prompt and response API. However, SPA will be applicable once they offer APIs to their logits and token embeddings.
>
> `The hyperparameter that controls the strength of the logit amplification is model and dataset -dependent. It needs to be tuned as it exhibits a clear maxima. This effect is well discussed in appendix J and the authors show that, while the parameter is strongly model-dependent, the method can still be effective with a fixed value across the benchmark datasets. Unfortunately, the authors do not compare directly in a table the fixed approach with the other SOTA methods. This would be desired, as the real question is whether the proposed approach beats SOTA for a deployable model (dataset-independent).`
>
> Thank you for the suggestion. We conducted an additional experiment to evaluate a fixed anchoring strength across different benchmarks. Specifically, we got this fixed value by averaging the tuned anchoring strengths across benchmarks for each model. In the table below, we update Table 2 by adding a new condition (SPA-fixed). It shows that SPA is effectively deployable in new scenarios once a reasonable value is set. We will add more discussion about the practical deployment of SPA in Appendix J.3.
>
> | Method          | Improvement |
> |----------------|:-------------:|
> | PASTA          | + 1.2        |
> | Self-Debugging | + 4.2        |
> | Self-Edit      | + 1.8        |
> | Self-Planning  | + 3.6        |
> | ReAct          | + 1.3        |
> | SPA-fixed      | + 5.0        |
> | SPA            | + 7.7        |

---

### Official Review · Reviewer_Nvi5 · 2025-03-12

**Overall Recommendation:** 4

**Summary:**

This work identifies attention dilution as a cause of code performance worsening as the context (generated code) increases. They subsequently propose a solution based on attention steering to upweight relevant tokens. This effectively shortens the effective context of the model and they show substantial performance (12.9%) improvement.

**Claims And Evidence:**

The consistent improvement is impressive especially if omega can be consistently selected without bootstrapping on a subset of problems.
The claims seem well substantiated.

**Essential References Not Discussed:**

Previous work has shown attention blocks tend to focus on the recent tokens but also on early tokens (corresponding to system prompt):

StreamingLLMs: https://arxiv.org/pdf/2309.17453

In settings where the model does indeed focus on the early tokens mostly, the dilution may not be of that much of a concern. However, it would be interesting if it was a code model specific phenomena.

Other work also looks at paired decoding capturing the differential signal based on the prompt:

context-aware decoding: https://arxiv.org/pdf/2305.14739

**Experimental Designs Or Analyses:**

It's not clear why exactly there's a 1.25X inference overhead. Is this simply a result of applying omega to the original embeddings? Figure 1  and eq 14 seems to indicate a parallel decoding. Is this included in the 1.25X overhead.

Line 175 come out of nowhere. Why is it that you only run this steering when there are failed test cases? If this consistently leads to improved performance it seems natural to always run with the steering. Additionally, this would implicitly avoid any errors introduced by incorrectly setting omega -- leading to upside only. In practice, you would be incurring at least 2x the overhead by having to generate and then run test cases, before rerunning. Is this rerunning included in the 1.27x overhead?

**Methods And Evaluation Criteria:**

The motivation of the proposed method is difficult to understand but boils down to a simple reweighing of the logits based on masking vs no masking of the prompt text (anchoring text).

**Other Comments Or Suggestions:**

It would be good to have the discussion on how to select omega in the main body of the text and move parts of the derivation to the appendix. Empirically, it seems the choice of omega is important.

**Other Strengths And Weaknesses:**

I worry that MBPP and HumanEval are in the memorization regime. It would support the argument to demonstrate the same trends on LiveCodeBench or other coding tasks to show this behavior also occurs when the model is operating on uncontaminated data.

I really like that the attention steering is a very general approach and should apply also to vision and other non-code language tasks. This initial work provides strong evidence to explore the impact in other domains.

**Questions For Authors:**

Can you provide intuition especially for the code domain why the last layer "has been shown to represent the most accurate attention distribution" (line 86)?

In what way is the attention being steered and not just the final logits? This may be clarified by clarifying Ei(X, ω).

**Relation To Broader Scientific Literature:**

This work proposes a paired decoding technique that's especially effective for coding applications. The derivation is missing some details but it's a neat approach to steering models to pay more attention to relevant code segments. This work relates well to localization for coding agents. In that literature, it's understood that the precision of localization has a large impact on the agent's performance. This work seems to indicate clever attention steering can reduce the effective context and implicitly does localization.

**Theoretical Claims:**

"For simplicity, we demonstrate by making the entire user prompt x as the anchored text." As figure 1 indicates the choice of the anchored text is important this simplification does not help and makes it more confusing as to what the Masking does. It seems that the approach is to mask out the anchored text and see the impact on the log probabilities.

Ei(X, ω) is not formally defined and so It's not clear in the derivation why "Scaling the semantics of X by ω times is not equivalent to multiplying X by ω.".

---

> ### Author Rebuttal · Authors · 2025-04-01
>
> `I worry that MBPP and HumanEval are in the memorization regime... `
>
> Thank you for the suggestion. We conducted a new experiment on LiveCodeBench (10/1/2024-2/1/2025). The results show that SPA remains effective.
>
> | Model                        | LiveCodeBench             |
> |-----------------------------|---------------------------|
> | Codegen-mono (350M)         | 1.1                       |
> | +SPA                        | 1.5 (+0.4) (36%↑)         |
> | DeepSeek-Coder (1.3B)       | 6.5                       |
> | +SPA                        | 9.2 (+2.7) (42%↑)         |
> | DeepSeek-Coder (6.7B)       | 7.8                       |
> | +SPA                        | 10.8 (+3.0) (38%↑)        |
> | CodeLlama (7B)              | 3.8                       |
> | +SPA                        | 4.0 (+0.2) (5%↑)          |
> | StarCoder2 (16B)            | 7.0                       |
> | +SPA                        | 8.2 (+1.2) (17%↑)         |
> | DeepSeek-Coder (33B)        | 11.9                      |
> | +SPA                        | 15.8 (+3.9) (33%↑)        |
>
> `It's not clear ... Is this rerunning included in the 1.27x overhead?`
>
> The inference overhead includes running test cases, the two forward passes, and the logit arithmetic operations. We did a further analysis of these different sources of overhead and found that the overhead for running test cases only incurs a very small overhead (0.1s on average) compared to the entire inference process (9.6s). Thus, only performing steering in case of test failures actually reduces the overhead. We did not perform a parallel decoding but we think this is a great idea and will discuss how this will further reduce the overhead in future work. Thank you!
>
> In our experiments, we use test cases in the benchmark instead of prompting the LLM to generate new test cases. Leveraging existing test cases is a common practice in code generation, such as [1] and [2]. But we agree that if there are no test cases available, prompting the LLM for test generation could be a worthwhile solution even with the additional overhead.
>
> [1] Chen et al. Teaching large language models to self-debug. ICLR 2024.
>
> [2] Zhang et al. Self-Edit: Fault-aware code editor for code generation. ACL 2023.
>
> `"For simplicity, we demonstrate by making the entire user prompt x as the anchored text."...`
>
> Sorry for the confusion. We initially intended to use anchoring the entire prompt as an example scenario to illustrate how the attention steering mechanism works in Section 3.3. We will change this writing and present the anchored text more generally and formally.
>
> `Ei(X, ω) is not formally defined so It's not clear why "Scaling the semantics of X by ω times is not equivalent to multiplying X by ω."`
>
> Ei(X,ω) represents the augmented embedding matrix of Ei (Line 142 after Eq. 5), where the semantic influence of X on the generated output is scaled by ω. Simply multiplying the embedding of X by ω does not simply improve the “**semantic influence**” of X since the embedding of X also encodes other non-semantic information such as the positional information. This is why we compute the difference between the logits when masking and unmasking X to cancel out noises and some of the non-semantic information. We will clarify this in the paper.
>
> `In what way is the attention being steered and not just the final logits?`
>
> As shown in Section 3.3., SPA can mathematically simulate attention steering via logit arithmetics. We chose not to directly modify self-attention since it is brittle and costly. For example, PASTA requires an expensive model profiling stage to identify usable attention headers to steer. Furthermore, only modifying self-attention layers does not account for the impact of other components like the feedforward layers and may cause an adversary effect given that self-attention and other components are trained together. As shown in Table 2, SPA is more computationally efficient and also achieves better performance than PASTA.
>
> `Can you provide intuition for the code domain why the last layer "represent the most accurate attention distribution"?`
>
> According to [3], while lower attention layers capture local dependencies such as the variable relationship within an expression, deeper layers capture more abstract representations with long-distance dependencies such as the control flow. Deeper layer attention distribution mirrors how humans understand programs by integrating information across the project. [4] confirmed this by showing attention distributions in the last layer produce the highest alignment between model and human.
>
> [3] Wan et al. What Do They Capture? A Structural Analysis of Pre-Trained Language Models for Source Code. ICSE 2022.
>
> [4] Kou et al. Do Large Language Models Pay Similar Attention Like Human Programmers When Generating Code? FSE 2024.
>
> `It would be good to move how to select omega in the main body.`
>
> Thank you for the suggestion. We will move Appendix J.1 to Section 5.

---

### Official Review · Reviewer_8eZp · 2025-03-18

**Overall Recommendation:** 3

**Summary:**

This paper identifies an attention dilution problem in code generation using LLMs, where models pay decreasing attention to the user prompt as more code tokens are generated. To address this issue, the authors propose Selective Prompt Anchoring (SPA), a model-agnostic approach that amplifies the contextual impact of user prompts during generation. SPA works by calculating the difference between logit distributions from original and masked versions of anchored text, then scaling this difference by a hyperparameter. Experiments across multiple benchmarks (HumanEval, MBPP, etc.) and models demonstrate consistent improvements, with Pass@1 increasing a bit.

**Claims And Evidence:**

The primary claims regarding attention dilution and SPA's effectiveness are well-supported. The authors provide persuasive empirical evidence demonstrating attention patterns across multiple models (Figures 2-4). The performance improvements are consistently observed across diverse models and benchmarks.

However, the claim that attention dilution is the reason for code generation errors could be better substantiated. While correlation between incorrect code and longer generation length is shown, this doesn't necessarily show causation, e.g, "code is longer" maybe just mean the question itself is harder.

**Essential References Not Discussed:**

N/A

**Experimental Designs Or Analyses:**

The experimental design is thorough with appropriate controls and ablations.

A limitation is the selection of "anchored text." This approach appears to be somewhat fixed and not "clever", and constrained to HumanEval-related tasks and may not generalize well to more complex programming scenarios (e.g. in some other code tasks, NL might not be that important compared to code") . While the authors explore various selection strategies in Section 5.4, a more intelligent approach for identifying optimal anchored text across diverse programming contexts would significantly strengthen the practical applicability of SPA.

**Methods And Evaluation Criteria:**

- The evaluation is comprehensive, using established benchmarks (HumanEval, MBPP, and variants) along with multilingual code generation (HumanEval-X) and more complex real-world tasks (BigCodeBench).
- The comparison against both attention steering (PASTA) and prompting methods (Self-Debugging, Self-Edit, etc.) provides a well-rounded assessment.
- inference time is also analyzed, but maybe better to discuss a little bit about potential memory overhead or precomputation requirements

**Other Comments Or Suggestions:**

See above comments

**Other Strengths And Weaknesses:**

See above comments

**Questions For Authors:**

- My understanding is that the SPA method amplifies the attention of specific tokens of the prompt. Now, would this approach, to some extent, change the model’s original behavior, and like, possibly affecting outputs that were initially correct? It seems plausible that it could introduce some bias, but I suppose the actual impact would depend on the context and how the method is applied.
- How would SPA perform with more dynamic anchored text selection strategies that change during the generation process?
- The paper shows that SPA is particularly effective for longer prompts. Would it be suitable for more complex programming tasks like super long prompts (even whole repos) and sometimes, random prompt (hard to define which parts are more important)?

**Relation To Broader Scientific Literature:**

- The paper effectively situates SPA within both code generation and attention steering literature. The connection to the psychological concept of the "anchoring effect" provides an interesting interdisciplinary perspective.

- The work advances upon prior attention steering methods like TOAST and PASTA by offering a more model-agnostic approach with lower computational overhead. It also contributes to the growing body of research on training-free methods for improving LLM performance.

- A notable constraint is that SPA appears most effective for code generation tasks specifically. The paper's own experiments in Section 5.6 show limited effectiveness on other generative tasks (e.g., MMLU), suggesting task-specificity that restricts its broader applicability in the LLM literature.

**Theoretical Claims:**

The mathematical derivation of SPA in Section 3 is generally sound.

---

> ### Author Rebuttal · Authors · 2025-04-01
>
> `While correlation between incorrect code and longer generation length is shown, this doesn't necessarily show causation, e.g, "code is longer" maybe just mean the question itself is harder.`
>
> This is a good point! We investigated this using LiveCodeBench, which provides the difficulty level for each task. For tasks with the same difficulty level, we still observed a significant difference between correct solutions and incorrect solutions. We appreciate your insightful feedback and will include this new result in the appendix.
>
> |         |   Easy   |  Medium  |   Hard   |
> |---------|:--------:|:--------:|:--------:|
> | Passed  |   294    |   475    |   400    |
> | Failed  |   418    |   664    |   784    |
>
>
>
> `inference time is also analyzed, but maybe better to discuss a little bit about potential memory overhead or precomputation requirements`
>
> There is no precomputation requirement. For memory overhead, SPA needs to store the logits computed from masked prompt embeddings. Theoretically, the extra memory is equal to vocabulary_size * token_embedding_dimension * logit_size.
>
> Suppose:
>
> - vocabulary_size = 50,000
> - token_embedding_dimension = 4,096
> - logit_size = 2 bytes
>
> The logit overhead will be 50,000 * 4,096 * 2 ≈ 390 MB.
>
> In practice, this memory overhead can be further reduced because low-ranked tokens do not contribute significantly. We can only augment the logits of a few top tokens. For example, if we consider the top 100 logits, the overhead will dramatically drop to 800 KB.
>
> We will discuss this in the paper.
>
> `My understanding is that the SPA method amplifies the attention of specific tokens of the prompt. Now, would this approach, to some extent, change the model’s original behavior, and like, possibly affecting outputs that were initially correct? It seems plausible that it could introduce some bias, but I suppose the actual impact would depend on the context and how the method is applied.`
>
> Indeed, as shown in Figure 6, different anchoring strengths ω of SPA would have different impacts. If the strength is too low, the performance improvement is low; if it is too high, the LLM starts becoming biased, leading to a decline in performance. Therefore, the actual impact depends on how we set this hyperparameter. Fortunately, we observe that it is simple to tune a balanced value where performance significantly improves. Please check more details in Appendix J. We will clarify this in the paper.
>
> `A limitation is the selection of "anchored text." This approach appears to be somewhat fixed and not "clever", and constrained to HumanEval-related tasks and may not generalize well to more complex programming scenarios (e.g. in some other code tasks, NL might not be that important compared to code") . While the authors explore various selection strategies in Section 5.4, a more intelligent approach for identifying optimal anchored text across diverse programming contexts would significantly strengthen the practical applicability of SPA.`
>
> `How would SPA perform with more dynamic anchored text selection strategies that change during the generation process?`
>
> It is a very interesting future work to investigate how to intelligently select anchored text. One idea is to use LLMs to first select important words or phrases to anchor on before code generation. For tasks where NL may not be important compared to code (e.g., code translation), one idea is to use static code analysis to identify important code elements (e.g., function calls and variable names heavily used in the code).
>
> Furthermore, the anchoring strength, which determines the degree of attention steering, can also change dynamically at different steps. One idea is to develop a method to calculate the relevance of words and phrases in the user prompt to each decoding step. Based on the relevance scores, SPA can dynamically assign higher values to more relevant contexts while assigning lower values to less relevant ones.
>
> We will discuss these ideas as future work in the paper.
>
> `The paper shows that SPA is particularly effective for longer prompts. Would it be suitable for more complex programming tasks like super long prompts (even whole repos) and sometimes, random prompts (hard to define which parts are more important)?`
>
> SPA is suitable for super long prompts since specific instructions like "do not use lambda expressions in the generated code" are likely to be buried in the long prompt and not followed by the LLM. SPA will help improve the influence of such specific instructions.
>
> In fact, it may be more helpful to use SPA in a self-improving pipeline for fairly complex tasks. Based on the errors of the initially generated code, we prompt the LLM to identify which instructions or requirements are not followed in the prompt and then use SPA to improve the influence of these instructions/requirements.

---

### Decision · Program_Chairs · 2025-05-01

**Decision:**

Accept (poster)

**Comment:**

The reviewers broadly agreed that this work presents an interesting approach that significantly improves the performance of the models it is applied to. The motivation is well-supported by empirical evidence and the writing is clear. While they note that the approach is somewhat invasive and may have potential, unobserved downsides, the gains in performance and thorough evaluation clearly support its net positive value. Given this, we recommend its acceptance.